# Sensitivity of the radiative forcing by stratospheric sulfur geoengineering to the amount and strategy of the SO$_2$ injection studied with the LMDZ-S3A model

Christoph Kleinschmitt[1,2], Olivier Boucher[3], and Ulrich Platt[1]

[1]Institute of Environmental Physics, Heidelberg University, Im Neuenheimer Feld 229, 69120 Heidelberg, Germany
[2]Laboratoire de Météorologie Dynamique, Institut Pierre-Simon Laplace, CNRS / UPMC / Sorbonne Université, 4 Place Jussieu, 75252 Paris CEDEX 05, France
[3]Institut Pierre-Simon Laplace, CNRS / UPMC / Sorbonne Université, 4 Place Jussieu, 75252 Paris CEDEX 05, France

*Correspondence to:* Christoph Kleinschmitt (christoph.kleinschmitt@iup.uni-heidelberg.de)

**Abstract.** The enhancement of the stratospheric sulfate aerosol layer has been proposed as a method of geoengineering to abate global warming. Previous modelling studies found that stratospheric aerosol geoengineering (SAG) could effectively compensate the warming by greenhouse gases on the global scale, but also that the achievable cooling effect per sulfur mass unit, i.e. the forcing efficiency, decreases with increasing injection rate. In this study we use the atmospheric general circulation model LMDZ with the sectional aerosol module S3A to determine how the forcing efficiency depends on the injected amount of SO$_2$, the injection height and the spatio-temporal pattern of injection. We find that the forcing efficiency may decrease more drastically for larger SO$_2$ injections than previously estimated. As a result, the net instantaneous radiative forcing does not exceed the limit of –2 W m$^{-2}$ for continuous equatorial SO$_2$ injections and it decreases (in absolute value) for injection rates larger than 20 Tg S yr$^{-1}$. In contrast to other studies, the net radiative forcing in our experiments is fairly constant with injection height (in a range 17 to 23 km) for a given amount of SO$_2$ injected. Also spreading the SO$_2$ injections between 30°S and 30°N or injecting only seasonally from varying latitudes does not result in a significantly larger (i.e. more negative) radiative forcing. Other key characteristics of our simulations include a consequent stratospheric heating, caused by absorption of solar and infrared radiation by the aerosol, and changes in stratospheric dynamics, with a collapse of the quasi-biennial oscillation at larger injection rates, which has impacts on the resulting spatial aerosol distribution, size, and optical properties. But it has to be noted that the complexity and uncertainty of stratospheric processes cause considerable disagreement between different modelling studies of stratospheric aerosol geoengineering. This may be addressed through detailed model intercomparison activities, as observations to constrain the simulations of stratospheric aerosol geoengineering are not available and analogues (such as volcanic eruptions) are imperfect.

## 1   Introduction

Solar radiation management (SRM) has been proposed as a possible means to offset increasing Earth's temperature in response to the anthropogenic greenhouse effect should mitigation efforts be insufficient and/or come too late. Among SRM techniques, stratospheric aerosol geoengineering (SAG) has received a lot of attention, starting with Budyko (1977) and largely reinstated

by Crutzen (2006). A large number of studies have been published since then, whose objectives fall in different categories : i) understanding the physics and chemistry of sulfate stratospheric aerosols, their radiative impacts and other potential impacts on the stratosphere, e.g. on ozone (e.g., Tilmes et al., 2008; Rasch et al., 2008; Pierce et al., 2010; Pitari et al., 2014; Aquila et al., 2014), ii) optimizing in some sense the stratospheric aerosol injection in space and time or by using different aerosol types (e.g., Heckendorn et al., 2009; Niemeier et al., 2011; Ferraro et al., 2011; English et al., 2012; Niemeier and Timmreck, 2015) , and iii) estimating the climate response to such stratospheric aerosol geoengineering, and to which extent it compensates or not global warming expected from anthropogenic greenhouse effects (e.g., Robock et al., 2008; Tilmes et al., 2009; Ricke et al., 2010; Kravitz et al., 2013).

Despite an important research effort on SAG over the last 10 years, there remains a number of uncertainties and open questions. Recent reviews on the subject (e.g., Irvine et al., 2016; MacMartin et al., 2016; Visioni et al., 2017) highlighted the need for accurate stratospheric aerosol models. This is important indeed because the central idea of SAG is that stratospheric aerosols scatter sunlight back to space, thus cooling Earth, but they also absorb and emit infrared radiation, thus causing a warming influence. The net effect is sensitive to the aerosol spatial distribution and the aerosol microphysics, which in turn depend on an accurate representation of key processes in the stratosphere. In this study we make use of the new Sectional Stratospheric Sulfate Aerosol (S3A) module developed within the atmospheric general circulation model LMDZ, which is described in detail in Kleinschmitt et al. (2017).

We focus here on SRM using stratospheric sulfate aerosol (formed from injections of $SO_2$) because volcanic sulfate aerosol provides a natural analogue which largely confirms its capability to cool the Earth's surface. Other aerosol types have been proposed in the last years such as solid $TiO_2$, $Al_2O_3$, $CaCO_3$ or even diamond particles (Weisenstein et al., 2015; Dykema et al., 2016; Keith et al., 2016) because of their possibly more favourable optical and/or chemical properties. We do not explore these further in this article but our model could be adapted in the future to study other aerosol types or mixtures of different aerosol types.

One open question regarding sulfate SAG is how the net radiative forcing scales with the amount of aerosol precursor or aerosol injected. At large sulfuric acid concentrations, particles tend to grow larger through condensation and coagulation, resulting in a decreased mass scattering efficiency for solar (shortwave = SW) radiation, while the mass absorption efficiency for terrestrial (longwave = LW) radiation remains essentially unchanged. This results in a less negative net forcing efficiency (i.e., radiative forcing per unit of injection rate) as the injection rate increases. Niemeier and Timmreck (2015) have studied the relation between sulfur dioxide injection rate and global mean radiative forcing in a series of simulations and found that for a 10-fold increase in injection rate (from 5 to $50 \, \text{Tg} \, \text{S} \, \text{yr}^{-1}$), the net forcing efficiency is reduced by 50 %.

The competition between the (mostly negative) SW radiative forcing and (mostly positive) LW radiative forcing of stratospheric aerosols is an important issue, as both effects can be of similar magnitude. As the net forcing is the sum of these partly compensating terms which both have considerable uncertainty, the overall uncertainty of the net forcing is even larger. For this reason, we discuss SW, LW and net (SW+LW) forcings independently in this study. Here, we focus on instantaneous radiative forcings, but the radiative effects of rapid adjustments (leading to an effective radiative forcing) are discussed in a companion article (Boucher et al., 2017) in the context of combining stratospheric aerosol geoengineering and marine cloud brightening.

Another important parameter of SAG is the injection height for the aerosol or its precursors. In principle, one would expect the forcing efficiency to increase with injection height, because it would take more time for sedimentation to transport the aerosol below the tropopause. However sedimentation is not the only relevant process. Stratospheric dynamics also plays an important role in determining the aerosol spatial distribution and therefore its lifetime. Aerosol injected or formed in the lower stratosphere may be transported polewards and return into the troposphere quicker than aerosol injected in middle stratosphere because the lower branch of the meridional Brewer-Dobson circulation is more rapid than the higher branch. For aerosols injected in the tropics, the speed of the meridional transport also depends on the phase of the Quasi-Biennial Oscillation (QBO) (Punge et al., 2009). During the QBO easterly phase, the subtropical barrier is more effective and the tropical pipe is stronger, thus, the meridional transport is slower. In contrast, during the QBO westerly phase the subtropical barrier is less effective, the tropical pipe is shifted up, and the meridional transport is faster. However it could be that large aerosol injections feed back on stratospheric circulation and hence on the aerosol stratospheric lifetime (Aquila et al., 2014; Niemeier and Schmidt, 2017; Richter et al., 2017).

At the same time, the "effort" spent on SAG and thus the cost of SAG measures strongly depends on injection height, because conventional aircraft are not optimised for flying at high altitude. Therefore, the development of a special aircraft fleet or an alternative technology aimed at higher injections, if required, would result in significantly higher economic costs (McClellan et al., 2012). The costs as a function of injection height are nevertheless not well established, so it is not straightforward to decide if it is cheaper to inject a larger quantity at a lower altitude or a smaller quantity at a higher altitude. There are few studies that look at the economic cost of SAG. However, we may naïvely expect that the injection at several locations may be more expensive (because it would require more infrastructure and limit economies of scale) or less expensive (because it would be more effective to create radiative forcing, as MacMartin et al. (2017) found in a recent study). Nevertheless, the objective of this study is not to find the cheapest injection strategy but to investigate how the radiative forcing depends on the injected amount of sulfur, the injection height, the number of injection points, and the temporal pattern of the injection. In this context we also study the impact of SAG on stratospheric dynamics and to which extent the additional aerosol of stratospheric origin increases particle concentrations and acid deposition at the Earth's surface.

## 2 Model and simulations

### 2.1 Model description

In this study we use the atmospheric general circulation model LMDZ (Hourdin et al., 2006, 2013) with the new sectional stratospheric sulfate aerosol module S3A described in detail in Kleinschmitt et al. (2017).

In the present configuration, LMDZ is run with $96 \times 96$ grid-points on the horizontal – i.e., a resolution of $1.89°$ in latitude and $3.75°$ in longitude which is the same as for LMDZ5A (Hourdin et al., 2006, 2013) – and 79 layers on the vertical in order to improve the vertical resolution in the stratosphere. The pressure at the top of the model is $4\,\mathrm{Pa}$ (corresponding to roughly $80\,\mathrm{km}$ altitude). Although it is possible to couple LMDZ interactively to an ocean model to form the IPSL coupled atmosphere-ocean model, we decided here to only perform simulations with fixed sea surface temperatures (SST). This is because our focus is on

estimating radiative forcings and because this choice allows us to perform more and longer simulations for a given computational cost. It has to be highlighted that due to the fixed SST it is not possible to evaluate the overall impact of SAG on climate, but we can analyse instantaneous radiative effects of the aerosol as well as the fate of the sulfate particles in the troposphere. Rapid adjustments on the temperature, water vapour and cloud fields are discussed in a separate study (Boucher et al., 2017).

Special attention was paid to the simulation of stratospheric dynamics in the model in general, and the Brewer-Dobson circulation and the quasi-biennial oscillation (QBO) in particular (de la Cámara et al., 2016; Kleinschmitt et al., 2017). Since the relevance of certain physical processes varies between the troposphere and the stratosphere, it is important to diagnose the tropopause during the simulation. Therefore, LMDZ-S3A computes the tropopause pressure at every time step following the WMO definition based on the lapse rate (Reichler et al., 2003). The tropopause height is used to estimate the budget of
stratospheric aerosols.

S3A includes a representation of sulfate particles with dry radii between $1\,\mathrm{nm}$ and $3.3\,\mathrm{\mu m}$ discretised in 36 logarithmically spaced size bins, as well as the precursor gases OCS, $SO_2$ and $H_2SO_4$. The physical processes of nucleation, condensation, evaporation, coagulation and sedimentation are represented. It should be noted that we only consider coagulation of aerosol particles from Brownian diffusion (Jacobson et al., 1994) and do not account for the enhancement effect due to van der Waals
(vdW) forces when computing the coagulation kernels. It has been shown that van der Waals forces may enhance coagulation by 30% for the smallest particles but their importance decreases rapidly as particle size increases (Chan and Mozurkewich, 2001). Sensitivity studies performed by English et al. (2013) and Sekiya et al. (2016) simulating the 1991 eruption of Mount Pinatubo found that including inter-particle van der Waals forces increased the peak effective radius by $10\,\%$ and decreased stratospheric AOD and burden by $10\,\%$. A further sensitivity test performed in our model has shown that considering the van
der Waals coagulation term resulted in a relatively small shift in the particle size distribution towards larger sizes and a small decrease in the simulated peak global-mean stratospheric aerosol optical depth in the weeks following the Pinatubo eruption (Kleinschmitt et al., 2017). Given the large uncertainties on how to parametrise the vdW effect on coagulation, we decided not to consider this process in most of our experiments. However, in Sect. 3.5 we discuss the results of a sensitivity test including vdW coagulation enhancement under conditions of strong SAG.

An important feature of our model is the fact that aerosols are fully interactive with the radiative transfer scheme, thus allowing changes in atmospheric heating rates due to stratospheric aerosols. We showed in Kleinschmitt et al. (2017) that aerosol heating causes an upward motion that lifts the aerosol in a simulation of the Pinatubo eruption. Aerosol optical properties are computed from Mie theory using the simulated size distribution and a prescribed chemical composition consisting of 75% $H_2SO_4$. We compute spectrally-averaged optical properties for the 6 solar and 16 terrestrial wavebands of the radiative transfer model as
well as for the 550 nm and $10\,\mathrm{\mu m}$ wavelengths for the sake of analysing the results.

A limitation of our experimental setup is the assumed fixed chemical timescale for $SO_2$ to $H_2SO_4$ conversion. This neglects any feedbacks of OH depletion onto $SO_2$ oxidation rate and lifetime. While this effect has been shown to be important for exceptional eruptions such as that of Tambora (Bekki, 1995), we estimated that this effect only limits oxidation for a short period after an eruption of the size of Pinatubo (Kleinschmitt et al., 2017) with simulated maximum grid-box $SO_2$ mass mixing ratios
of 71 ppm (in daily mean) and 2.6 ppm (in monthly mean). We therefore expect this effect to be smaller for our continuous

**Table 1.** Description of the experiments performed in this study.

| Experiment | Description |
| --- | --- |
| CONTROL | Control experiment with background stratospheric sulfate aerosols |
| STANDARD | Continuous equatorial injection of $10 \, \mathrm{Tg \, S \, yr^{-1}}$ as $SO_2$ at $17 \pm 1 \, \mathrm{km}$ into one equatorial grid cell |
| $x$_TGS | Same as STANDARD, but with different injection rates, where $x = 2, 5, 20, 50 \, \mathrm{Tg \, S \, yr^{-1}}$ |
| $z$_KM | Same as STANDARD, but with different injection heights, where $z = 15, 19, 21, 23 \, \mathrm{km}$ |
| BROAD | Same as STANDARD, but injection distributed at 28 locations around the globe between $30°\mathrm{N}$ and $30°\mathrm{S}$ |
| SEASONAL | Same as STANDARD, but injections into one grid cell limited to two months of the year (at $5°\mathrm{N}$ in April and at $5°\mathrm{S}$ in October) |
| $x$_TGS, $E(0)/E(\infty)$ | Same as $x$_TGS experiments (with $x = 20, 50 \, \mathrm{Tg \, S \, yr^{-1}}$), but with coagulation enhanced by van der Waals forces (continuum regime factor $E(0)$ and kinetic regime factor $E(\infty)$, respectively) |
| NORAD | Same as STANDARD, but with radiatively non-interactive aerosol and an injection height of $21 \, \mathrm{km}$ |

SAG experiments where maximum $SO_2$ mixing ratios do not exceed $3.4 \, \mathrm{ppm}$ in daily mean (and $2.2 \, \mathrm{ppm}$ in monthly mean) for a continuous injection rate of $10 \, \mathrm{Tg \, S \, yr^{-1}}$ and $16 \, \mathrm{ppm}$ in daily mean ($8.4 \, \mathrm{ppm}$ in monthly mean) for $50 \, \mathrm{Tg \, S \, yr^{-1}}$. For continuous SAG schemes, where the aerosol layer and its properties reach an equilibrium after several years, we would expect oxidants to reach equilibrium concentrations (somewhat) lower than the climatological values. Assuming a climatological

chemical lifetime of $SO_2$ could therefore cause the model to overestimate $SO_2$ to $H_2SO_4$ conversion rates to a certain degree, favouring new particle formation over condensational growth.

## 2.2 Simulation setup

We define a STANDARD SAG scenario, upon which we then base sensitivity studies to estimate the role of certain parameters and features of the injection strategy. In this standard scenario, an amount of $10 \, \mathrm{Tg \, S \, yr^{-1}}$ is injected in the form of $SO_2$ gas

into one equatorial grid cell (at $1°\mathrm{N}$, $120°\mathrm{E}$). The equatorial injection takes advantage of the ascending branch of the Brewer-Dobson circulation (BDC) that contributes to lift the aerosol. The aerosol lifetime in the stratosphere in this configuration can be expected to be longer than for injection at higher latitudes, because of the poleward transport and subsequent removal through the BDC. Furthermore, the radiative forcing of the aerosol is generally larger (i.e., more negative) in the tropics because of the maximum in insolation. The injection is constant in time, i.e. we add the same amount of $SO_2$ at every single time step

of $30 \, \mathrm{min}$. Vertically the injection is distributed with a Gaussian profile in altitude centred at $17 \, \mathrm{km}$ with a standard deviation of $1 \, \mathrm{km}$ (similarly to our setup of the 1991 eruption of Mount Pinatubo discussed in Kleinschmitt et al. (2017)).

In a first series of model experiments the magnitude of the stratospheric injection is varied between 2 and $50 \, \mathrm{Tg \, S \, yr^{-1}}$, while all the other parameters are kept constant (experiments labelled $x$_TGS, see Table 1). In a second series the central injection height is varied between 15 and $21 \, \mathrm{km}$ (still with a standard deviation of $1 \, \mathrm{km}$) for a default injection rate of $10 \, \mathrm{Tg \, S \, yr^{-1}}$

(experiments labelled $z$_KM).

In addition to the sensitivity studies for injection magnitude and height, we performed experiments with different spatio-temporal injection patterns (BROAD and SEASONAL). In BROAD, the injection is spread equally in 28 locations with 7 latitudes (30°S, 20°S, 10°S, 1°N, 10°N, 20°N, 30°N) and 4 longitudes (120°W, 30°W, 60°E, 150°E). In SEASONAL, the injection is performed in the months of April and October only, switching from a location at 5°N to a location at 5°S, so as to always emit from the summer hemisphere. The STANDARD, BROAD and SEASONAL simulations therefore represent different injection strategies in terms of locations and logistics.

Each simulation is performed for 5 years and the results are shown for the $5^{th}$ year of the simulation, when the aerosol layer has clearly reached a steady state (with only little year to year variability) regarding burden, particle size and spatial distribution. The stratospheric model fields are generally averaged over all the grid cells above the tropopause (diagnosed at the grid cell and timestep resolution).

We define the radiative forcing as the all-sky aerosol direct radiative effect (ADE) at the top of the atmosphere (TOA) in $W\,m^{-2}$ for shortwave (SW), longwave (LW) and net radiation. At each time step the radiative forcings are computed as the difference of two calls of the radiation code (one including sulfate aerosol, one not). Therefore the radiative forcing does not include any rapid adjustments that take place in the land-atmosphere system, including changes in stratospheric temperature and water vapour. The differences between (instantaneous) radiative forcing and effective radiative forcing (ERF) that include rapid adjustments are discussed in a separate study (Boucher et al., 2017). It should also be noted that the change in stratospheric water vapour may also feedback onto stratospheric chemistry and $SO_2$ oxidation, which is not accounted for in our simulations. We have to point out that the presented radiative forcing values from the SAG model experiments include a small contribution from the background sulfate aerosol, which we do not subtract for simplicity. In the control experiment, the background sulfate aerosol has a global and annual mean net instantaneous radiative forcing of –0.04 $W\,m^{-2}$ (–0.05 $W\,m^{-2}$ in the SW and +0.01 $W\,m^{-2}$ in the LW).

In order to get a better understanding of the relation between forcing efficiency (in $W\,m^{-2}$ / $Tg\,S\,yr^{-1}$) and injection rate magnitude, we decompose it in the following way:

$$\text{forcing efficiency} = \frac{\text{radiative forcing}}{\text{injection rate}} = \frac{\text{radiative forcing}}{\text{AOD}(550\,nm)} \cdot \frac{\text{AOD}(550\,nm)}{\text{burden}} \cdot \frac{\text{burden}}{\text{injection rate}} \tag{1}$$

The first term (radiative forcing/AOD(550 nm) in $W\,m^{-2}$ / unit AOD) reflects how a certain optical depth translates into a radiative forcing and is related to the model's radiative transfer code, to the distribution of the stratospheric aerosols in space and time, but also to particle size (since for a given AOD(550 nm), larger particles have a larger LW forcing than small particles). The second term (AOD(550 nm)/burden in unit AOD / $Tg\,S$) shows how effectively the particles extinguish visible light and is related to particle size (and hence to the model's aerosol microphysics). The third term (burden/injection rate in $Tg\,S$ / $Tg\,S\,yr^{-1}$ = yr) is equal to the effective lifetime of the injected sulfur in the stratosphere and therefore strongly depends on the modelled transport and removal processes. It should be noted that this decomposition is similar to the one used by Schulz et al. (2006) but is applied here to stratospheric aerosols.

## 3 Simulation results

### 3.1 Results from the STANDARD experiment

Figure 1 shows the stratospheric burdens and the fluxes of the represented sulfur species for the STANDARD SAG simulation with an equatorial injection of $10\,\mathrm{Tg\,S\,yr^{-1}}$ at $17\,\mathrm{km}$ altitude. The injected $SO_2$ is converted to $H_2SO_4$ after a mean lifetime of

41 days, while a fraction of roughly $4\,\%$ leaves the atmosphere without being converted (mostly because it is injected below the tropopause and rapidly removed from the troposphere through wet scavenging). Conversion of OCS to $SO_2$ in the stratosphere (after a mean lifetime of 8 years) adds only $0.3\,\%$ to the injected sulfur mass. Therefore the stratospheric sulfur budget can be considered as essentially anthropogenic in this scenario. $H_2SO_4$ vapour has a lifetime of 2 days, with the nucleation and condensation processes converting $H_2SO_4$ vapour to the particulate phase. Thus $85\,\%$ of the stratospheric sulfur exists in the

form of $H_2SO_4$ aerosols which has a mean lifetime in the stratosphere of about 10 months.

The spatial distribution of the sulfate aerosol mass mixing ratio (in $\mathrm{\mu g\,S\,kg\,air^{-1}}$) in Fig. 2 shows a pronounced maximum in the tropical stratosphere around $21\pm4\,\mathrm{km}$ altitude, i.e. well above the injection zone at $17\pm1\,\mathrm{km}$. The sulfate concentration (in $\mathrm{\mu g\,S\,m^{-3}}$, not shown) peaks at about $19\pm3\,\mathrm{km}$, i.e. above the injection height but below the peak of the sulfate mass mixing ratio. Outside the tropics the sulfate mixing ratio is much lower, characteristic of a relatively strong tropical transport barrier,

and decreases further towards the poles. The spatial distribution of the injected $SO_2$ (not shown) is much more confined to the latitudinal band corresponding to the injection zone although it expands vertically a few $\mathrm{km}$ above the injection altitude. This is because the injected $SO_2$ is almost completely converted to sulfuric acid gas before significant latitudinal transport to higher latitudes can take place. The mixing ratio of sulfuric acid vapour (not shown either) is very low in the lower stratosphere due to its rapid consumption through particle formation and growth, but it increases above $30\,\mathrm{km}$, where the aerosol starts to

evaporate.

The absorption of SW and LW radiation by the aerosol can heat up the surrounding air considerably. The temperature anomaly in the STANDARD SAG scenario with respect to the CONTROL run is shown in Fig. 3. It reaches up to $16\,\mathrm{K}$ in the lower tropical stratosphere and decreases with increasing altitude and latitude. In the upper troposphere the heating is weaker due to the low particle concentrations, but it can reach up to $3\,\mathrm{K}$ just below the tropopause. This heating is significant and contributes to

the lifting of the injected $SO_2$ and the aerosol. We evaluated this heating for the Pinatubo simulation performed in Kleinschmitt et al. (2017) and found that the model may overestimate the temperature anomaly by a factor of 2 relative to MSU channel 4 observations. One possible reason for the discrepancy was that we prescribed $O_3$ to its climatological values, whereas $O_3$ mixing ratios are known to have decreased in large parts of the stratosphere after Pinatubo, thus decreasing absorption of SW radiation. However, $O_3$ mixing ratios may also increase in the tropics due to the hydrolysis of $N_2O_5$ (Fahey et al., 1993).

Tilmes et al. (2009) also found considerable stratospheric ozone depletion in model experiments of SAG. Therefore, the effects of SAG presented in this study that critically depend on the radiative heating may not be as important in reality and should be studied with other models as well.

The tropopause itself, which is diagnosed within the model following the WMO definition based on the lapse-rate, descends by about $1\,\mathrm{km}$ in the tropics due to the heating of the lower stratosphere and the upper troposphere. Santer et al. (2003) studied the

contribution of various forcings to observed tropopause height changes. They showed that after the Mount Pinatubo eruption a tropopause descent of 5–10 hPa (roughly 0.2–0.4 km) was observed. This was probably caused by a heating of the stratosphere and a parallel cooling of the troposphere by the volcanic sulfate aerosol. But the underlying trend over the last decades is an increase in tropopause height caused by well-mixed greenhouse gases, which heat the troposphere and cool the stratosphere.

Hence, this observed increase due to global warming might be compensated (or even overcompensated) by the effect of SAG. But it should also be noted that our simulations are based on fixed SST, while an interactive ocean might have an impact on temperature changes and therefore on the tropopause height.

The spatial distribution of particle size (i.e., effective radius $r_{eff}$) is shown in Fig. 4. With effective radii up to 0.5 μm the particles are largest above the injection region where the sulfate concentrations are the largest. The region with the largest

particles descends towards higher latitudes due to ongoing particle growth and sedimentation during the meridional transport through the BDC.

## 3.2 Sensitivity to injection magnitude

The relation between the magnitude of the stratospheric sulfur injections and the resulting net radiative forcing is shown in Fig. 5. The most surprising observation is that the forcing does not exceed the $1.9 \, \mathrm{W \, m^{-2}}$, which is reached in the 20_TGS

experiment, but decreases for an even larger injection rate of $50 \, \mathrm{Tg \, S \, yr^{-1}}$. This is because the SW forcing does not increase as strongly with injection magnitude as the LW forcing (with opposite sign), due to the increase in particle size.

In order to illustrate the non-linear impact of the injection magnitude on the spatial distribution of the stratospheric aerosol, we plotted the sulfate aerosol mass mixing ratio (MMR) normalised by the value in the STANDARD experiment and the ratio of the injection rates in Fig. 6. For a linear scaling of MMR with injection rate, this ratio would be unity everywhere

(so the base-10 logarithm of the ratio –which is the quantity plotted– would be zero). In our model simulations, for a small injection of $2 \, \mathrm{Tg \, S \, yr^{-1}}$, the ratio is well below 1 above and below the injection region, while it is above 1 in the polar stratosphere. For a large injection of $50 \, \mathrm{Tg \, S \, yr^{-1}}$, the ratio shows the opposite pattern (apart from the zone immediately above the injection region). From this we can conclude that in relative terms with increasing magnitude of the injection rate 1) more sulfur accumulates below the injection region because of increasing sedimentation of larger particles and 2) less sulfur

reaches the high latitudes because of the induced changes in stratospheric dynamics (discussed in Sect. 3.7) and/or because of faster sedimentation during the meridional transport. The sulfate MMR above the injection region is largest (in relative terms) in the 10 and $20 \, \mathrm{Tg \, S \, yr^{-1}}$ scenario, probably because in these cases the updraft by aerosol heating is strong enough to lift the particles while the sedimentation is still relatively slow.

The mean forcing efficiency and its decomposition described above for the simulations with varying injection magnitude are

shown in Fig. 7. The forcing efficiency in the SW decreases by roughly 50 % between 2 and $50 \, \mathrm{Tg \, S \, yr^{-1}}$, while in the LW it stays rather constant. As a result, the net forcing efficiency decreases dramatically for larger injection rates (by 94 % between 2 and $50 \, \mathrm{Tg \, S \, yr^{-1}}$). Even the absolute forcing decreases from 20 to $50 \, \mathrm{Tg \, S \, yr^{-1}}$ (see Fig. 5). The decomposition shows that this decrease is mainly due to a decreasing net forcing/AOD (by 87 % between 2 and $50 \, \mathrm{Tg \, S \, yr^{-1}}$ injection rates), which itself is caused by an increasing LW forcing/AOD combined with a rather constant SW forcing/AOD. The AOD/burden also decreases

by more than 50 % due to larger particles (see Fig. 8). Despite this finding, the lifetime of the particles does not clearly decrease for larger injection magnitude. This may be explained by the superposition of two opposed effects: increasing sedimentation velocity and increasing updraft through heating of the air. Both can be seen in Fig. 6a, where the sulfate mixing ratio increases disproportionately with the injection rate above the injection region (due to updraft) and below it (due to sedimentation). But for the largest injection rate simulated, enhanced sedimentation starts to dominate (see Fig. 6b) so that the particle lifetime can be expected to decrease further for even larger injections.

Figure 8 shows the impact of the injection magnitude on particle size. The global mean effective radius (computed for all the aerosol above the tropopause) almost triples between 2 and $50\,\mathrm{Tg\,S\,yr^{-1}}$. The mass size distribution shown in Fig. 9 mainly differs in the size range above $0.1\,\mu\mathrm{m}$ such that the mode radius grows and is shifted towards larger radius with increasing injection rate. This explains the increase in LW forcing and therefore the decreasing forcing efficiency.

**Comparison with results from Niemeier and Timmreck (2015)**

Niemeier and Timmreck (2015) performed a similar sensitivity study for the SAG injection rate with the ECHAM model using a modal aerosol module. Their results deviate from ours in that the absolute net forcing increases monotonically with injection rate up to $100\,\mathrm{Tg\,S\,yr^{-1}}$, while the forcing efficiency decreases only moderately (see crosses in Fig. 7).

One important difference in the results of the two models is the evolution of particle size with increasing injection rate. Fig. 8 shows that in LMDZ-S3A the particles are smaller than in ECHAM for small injections and larger for large injections. Therefore the positive LW forcing increases more in LMDZ-S3A, resulting in a lower net forcing efficiency. This difference in particle growth may partly be caused by differences in aerosol microphysics (modal versus bin scheme) and/or differences in meridional transport between the two models. The transport barrier is quite weak in the ECHAM version with 39 vertical levels (without a generated QBO) used by Niemeier and Timmreck (2015) and transport is hardly affected by the aerosol, while in LMDZ-S3A the barrier is strengthened for larger injection rates because of the impact of the radiative heating on the stratospheric dynamics. Therefore the aerosol plume is even more confined for larger injections, leading to enhanced particle growth. In a recent study, Niemeier and Schmidt (2017) used a version of ECHAM with higher vertical resolution (L90) and more realistic stratospheric dynamics and also found that heating by aerosols slows down the meridional transport, making larger injections even less efficient.

We also found that the maximum extinction coefficient (in $\mathrm{km^{-1}}$) in both the SW and the LW bands is about a factor of 2 larger in LMDZ-S3A for a given injection rate. This causes a stratospheric heating which is also about twice as strong as in ECHAM. The stronger heating may explain why aerosol lofting plays a more important role in LMDZ-S3A, such that increasing sedimentation can be partly compensated (resulting in a longer aerosol lifetime). The difference in extinction between the models might be due to different particle size distributions, different computations of the aerosol optical properties or due to differences in the radiative transfer scheme.

Another difference may also partly explain the differing model results: In ECHAM, the global mean SW forcing/AOD increases for larger injections, keeping the net forcing/AOD rather constant. In contrast, in LMDZ-S3A the SW forcing/AOD

is rather independent of the injection rate, which in combination with increasing LW forcing/AOD causes the net forcing/AOD to decrease drastically. It remains unclear why the SW forcing/AOD increases with injection rate in ECHAM.

Kravitz et al. (2017) recently published a modelling study of SAG with $SO_2$ injection rates of up to $25\,\mathrm{Tg\,S\,yr^{-1}}$ and net forcing efficiency results similar to Niemeier and Timmreck (2015). But just as in our experiments, their individual SW and

LW forcing efficiencies were considerably larger than those found by Niemeier and Timmreck (2015), indicating differences in the aerosol's radiative effects between the models.

## 3.3    Sensitivity to injection height

Earlier model studies found a strong dependence of the forcing efficiency on the injection height, therefore we conducted a series of simulations with different injection heights (but all with the same injection rate of $10\,\mathrm{Tg\,S\,yr^{-1}}$).

Figure 10 shows that both the SW and the LW forcings increase with increasing injection height, but the net forcing (sum of SW and LW components) is almost completely independent of the injection height. Considering again the decomposition of the forcing efficiency given in Eq. 1, it appears that the net forcing/AOD decreases for higher injections, implying a less optimal size and spatial distribution of the aerosol. The AOD/burden ratio decreases as well due to larger particles (see Fig. 11), while the lifetime increases, as one would expect. There are various reasons for a shorter aerosol lifetime in the stratosphere

for lower injection height. First, a larger fraction of the sulfur is injected below the tropopause, so most of it does not enter the stratosphere at all and is rapidly removed from the troposphere. In the 15_KM experiment 72 % of the sulfur are injected below the tropopause, while in the 17_KM (STANDARD) experiment it is only 4 %. Second, in the lower stratosphere the tropical meridional transport barrier is less pronounced than above $20\,\mathrm{km}$. Therefore the sulfur is transported more effectively to higher latitudes, where it leaves the stratosphere again.

The lifetime increases for higher injections, but the mean effective particle radius increases as well. Particles might grow larger because the aerosol does not spread as rapidly as in the lower stratosphere and the larger local concentrations favour coagulation and condensation. Additionally, in the case of higher injections the mean stratospheric size distribution contains more particles which had more time to grow due to their longer lifetime.

The effects of longer lifetime and less optimal optical properties (larger particles) just cancel out each other, such that the overall

radiative forcing does not increase with injection height. This is in agreement with a recent study by Tilmes et al. (2017), who also found that for large equatorial $SO_2$ injections the effectiveness does not increase for injections at higher altitude. As higher injections can be expected to be technically challenging, the effort to produce a certain forcing even increases. Therefore our modelling results imply that it would not be worthwhile to inject the aerosol at altitudes higher than $19\,\mathrm{km}$. However the simulations show that injections at less than $17\,\mathrm{km}$ altitude would probably be very inefficient, because they result in a small

stratospheric aerosol burden.

The particle size, i.e. the effective radius, shown in Fig. 11 appears to approach a saturation level of slightly below $0.5\,\mathrm{\mu m}$ with increasing injection height. Only the aerosol mass on the lower end of the size range decreases further (see Fig. 12), probably because of faster coagulation. The particle concentration in all the size bins decreases when going from the tropics to mid and high latitudes, except for the 15_KM experiment, where the concentration of particles around $0.1\,\mathrm{\mu m}$ slightly increases

**Table 2.** Global mean quantities for experiments with $10\,\mathrm{Tg\,S\,yr^{-1}}$ injection rate but different spatio-temporal injection patterns.

| Experiment | Radiative Forcing (W m$^{-2}$) | | | AOD at 550 nm | $r_{\mathrm{eff}}$ (µm) | Aerosol Burden (Tg S) |
| --- | --- | --- | --- | --- | --- | --- |
| | SW | LW | net | | | |
| STANDARD | –4.6 | +3.1 | –1.5 | 0.19 | 0.37 | 8.1 |
| BROAD | –3.7 | +2.2 | –1.5 | 0.16 | 0.30 | 5.7 |
| SEASONAL | –4.8 | +3.1 | –1.6 | 0.20 | 0.38 | 8.4 |

towards mid latitudes. This may be due to relatively fast meridional transport of the aerosol through the lower branch of the BDC in this case.

## 3.4 Sensitivity to spatio-temporal injection pattern

Previous model studies (e.g., Niemeier and Timmreck, 2015) indicate that the choice, when and where to inject a given mass of SO$_2$, can affect the forcing efficiency considerably. Therefore we simulated two additional scenarios with an injection rate of $10\,\mathrm{Tg\,S\,yr^{-1}}$, but with injection patterns differing in time and space.

In the BROAD scenario, in which the sulfur mass is distributed over 28 points covering a larger area, the effective particle radius $r_{\mathrm{eff}}$ is almost 20 % below the one in the STANDARD scenario with injection into one equatorial grid box. However, the net forcing is almost equal in the two experiments, because the more favourable optical properties in the BROAD simulation are compensated by a smaller stratospheric aerosol burden. One reason for the smaller burden (and shorter lifetime) is the shorter average travel time from the injection region (30°N to 30°S) to mid/high latitudes where the aerosol is removed from the stratosphere. Another cause is that most of the aerosol remains below 20 km altitude, because it is not lifted by the BDC at the Equator and the more distributed injection pattern produces smaller radiative heating rates and therefore less updraft than in the STANDARD scenario. The AOD at 550 nm is much smaller in the tropics, but larger at mid latitudes.

In the SEASONAL scenario, where the sulfur mass is wholly injected during only two months of the year, the particles are slightly larger than in the STANDARD scenario. But because the burden is also a bit larger, the resulting net forcing is about 10 % larger than in the STANDARD scenario. The larger burden may result from a stronger updraft (due to larger aerosol concentrations in the short injection periods) that transports the aerosol to higher altitudes. This small increase in forcing efficiency seems to contradict a recent study by Laakso et al. (2017) who did not find a larger SW radiative forcing for seasonally varying injection scenarios. But it may be that the increase that we find is only caused by the stronger updraft / longer lifetime and not by the seasonal variations.

## 3.5 Sensitivity to the van der Waals coagulation enhancement factor

As argued in Sect. 2.1, the used model configuration of LMDZ-S3A does not include the enhancement of coagulation by van der Waals (vdW) forces, because sensitivity tests under Pinatubo conditions (Kleinschmitt et al., 2017) did not clearly improve

**Table 3.** Global mean quantities for experiments with and without van der Waals (vdW) enhancement of coagulation at injection rates of 20 and 50 Tg S yr$^{-1}$, respectively. The enhancement factor $E(\infty)$ for the kinetic regime is generally larger than the factor $E(0)$ for the continuum regime. Uniformly enhancing coagulation by $E(\infty)$ can therefore be seen as an upper limit of the impact of vdW forces.

| Experiment | Radiative Forcing (W m$^{-2}$) | | | AOD at 550 nm | $r_{eff}$ (µm) | Aerosol Burden (Tg S) |
|---|---|---|---|---|---|---|
| | SW | LW | net | | | |
| 20_TGS, no vdW | –8.1 | +6.3 | –1.9 | 0.33 | 0.46 | 17.1 |
| 20_TGS, $E(0)$ | –7.7 | +6.0 | –1.7 | 0.31 | 0.47 | 16.2 |
| 20_TGS, $E(\infty)$ | –6.4 | +5.5 | –0.9 | 0.24 | 0.52 | 14.0 |
| 50_TGS, no vdW | –14.8 | +13.7 | –1.0 | 0.62 | 0.60 | 41.1 |
| 50_TGS, $E(0)$ | –14.1 | +13.5 | –0.6 | 0.57 | 0.63 | 40.0 |
| 50_TGS, $E(\infty)$ | –11.9 | +12.2 | +0.4 | 0.45 | 0.67 | 34.4 |

the agreement with aerosol observations and there are large uncertainties on how this effect should be parametrised. In order to test if neglecting vdW forces is also justified under conditions of strong SAG, we performed additional sensitivity tests by repeating the 20_TGS and the 50_TGS experiments, once with the continuum regime vdW enhancement factor $E(0)$ and once with the (generally larger) kinetic regime vdW enhancement factor $E(\infty)$ from Chan and Mozurkewich (2001). The actual

enhancement factor for stratospheric conditions can be expected to lie in between these two cases.

Figure 13 shows that including a vdW enhancement factor causes a small shift in the aerosol size distribution towards larger radii. The effect is stronger for larger injection rates and for the larger enhancement factor $E(\infty)$, so that e.g. in the 50_TGS experiment with $E(\infty)$ the concentration of the largest particles almost doubles compared to the experiment without vdW enhancement. Table 3 lists the global and annual mean values of relevant quantities, revealing that the shift towards larger

particle size by the vdW enhancement results in a reduction of the stratospheric aerosol burden, the AOD at 550 nm, and a significant change in the radiative forcing. Since the SW forcing decreases more strongly than the LW forcing (in absolute values), the net radiative forcing becomes considerably less negative and can even become positive in the 50_TGS experiment with $E(\infty)$. We may therefore conclude that the strong decrease in forcing efficiency for larger injection rates observed in the experiments without vdW enhancement is probably underestimated if we were to account fully for the effect of vdW forces on

coagulation. But we should note that the exact impact of vdW forces on coagulation is not very well determined from theory and laboratory experiments and that the implementation in our model is only tentative.

### 3.6  Effect of radiatively interactive aerosol

We now discuss the impact of having radiatively interactive aerosols in our model simulations. Indeed the heating of the stratosphere and the upper troposphere described above can be expected to have a considerable impact on the atmospheric

dynamics and thereby on the distribution and evolution of the aerosol distribution itself. In order to quantify this impact, we performed an SAG simulation called NORAD in which the aerosol does not interact with radiation (only the instantaneous

radiative forcing is computed from a double radiation call but the model integration is performed with stratospheric aerosols that are invisible to the radiation). In this scenario, we chose a relatively high injection height of $21\,\mathrm{km}$, because without any aerosol-induced heating an important factor for the vertical transport from the tropical tropopause layer (TTL) to the stratosphere is missing.

The resulting vertical distributions of the aerosol extinction coefficient with and without radiative interaction are shown in Fig. 14. The heating causes the aerosol to rise higher and spread more meridionally, as it was already described by Aquila et al. (2012) for model simulations of the 1991 Mount Pinatubo eruption. This may also be related to the changes in stratospheric dynamics described below. The spatial distribution in the NORAD (at $21\,\mathrm{km}$) experiment is closer to the one in the 17_KM or 19_KM experiments, but with a more pronounced maximum in extinction coefficient. Due to the more confined spatial

distribution in the NORAD simulation, local concentrations of sulfuric acid and particles are larger, causing the particles to grow larger through condensation and coagulation. Many of the larger particles leave the stratosphere through sedimentation already in the tropics, so that the stratospheric aerosol burden in the NORAD experiment is smaller than in the 21_KM, 19_KM and even 17_KM experiments. Therefore, the resulting global mean net radiative forcing is significantly smaller in the NORAD experiment ($-0.9\,\mathrm{W\,m^{-2}}$) than in the 21_KM simulation with radiatively interactive aerosols ($-1.4\,\mathrm{W\,m^{-2}}$).

## 3.7   Impact on the QBO

The locally very strong heating of the lower stratosphere and upper troposphere due to the aerosol-radiation interactions (up to $16\,\mathrm{K}$ for the standard SAG scenario relative to the control run) can be expected to have a considerable impact on atmospheric dynamics. The quasi-biennial oscillation (QBO) in the equatorial stratosphere is an important dynamical feature of the stratosphere. It consists of easterly and westerly winds alternating with a period of approximately 28 months. The phase of the QBO

is known to affect the poleward transport of trace gases and aerosols in the stratosphere (Trepte and Hitchman, 1992).
Indeed Fig. 15 shows that the QBO is strongly affected by the sulfur aerosol injection, in agreement with a previous study by Aquila et al. (2014). While the QBO period in the spin up and CONTROL simulations including only stratospheric background aerosol is approximately 28 months (varying between 24 and 32 months as the background aerosol layer builds up), it increases significantly already for the smallest simulated injection rate of $2\,\mathrm{Tg\,S\,yr^{-1}}$. In the $5\,\mathrm{Tg\,S\,yr^{-1}}$ scenario the periodicity disap-

pears, while easterly winds dominate below $25\,\mathrm{km}$ and westerly winds dominate above. For the scenarios with $10\,\mathrm{Tg\,S\,yr^{-1}}$ and more, the direction of the propagation inverts from downward to upward, but with a period of approximately one year and with smaller amplitude.

    It has to be noted that the response of the QBO to SAG may be less severe when using interactive chemistry (Richter et al., 2017), probably due to ozone depletion resulting in a smaller stratospheric temperature anomaly.

## 3.8   Sulfate impact at the Earth's surface

The deposition of acids at the land or ocean surface can be harmful to ecosystems. Although the additional input of sulfur species to the Earth system is (at least for the lower emission scenarios) not large compared to the already existing anthropogenic and natural inputs (of about $136\,\mathrm{Tg\,S\,yr^{-1}}$ (Kravitz et al., 2009)) sulfate SAG constitutes an additional and intentional

sulfur input. Therefore, it is important to study the deposition fluxes of sulfuric acid due to SAG. Before their deposition sulfate particles near the surface can also have a negative impact on human health by increasing particle concentrations in ambient air ($PM_{2.5}$).

Due to the relatively short lifetime of the sulfate particles in the troposphere, the relevance and relative importance of the various processes at play are different than in the stratosphere. Therefore, we decided not to activate the microphysical processes of coagulation, nucleation, condensation and evaporation below the tropopause. Thus, the particles do not grow or shrink any more, but they are removed via wet and dry deposition (both independent of the particle size) and sedimentation.

However, our model setup makes it possible to study in first approximation the contribution of SAG to increase particle concentration at the Earth's surface (i.e., in the lowest atmospheric model layer) as well as the deposition rate of aerosols to the ground or sea surface.

The deposition rate of sulfate aerosol coming from the stratosphere to the surface in the STANDARD scenario is shown in Fig. 16. It is largest in the low and mid-latitudes and over the ocean, where it reaches up to $82\,\mathrm{mg\,S\,m^{-2}\,yr^{-1}}$. This spatial distribution is similar to the one shown by Kravitz et al. (2009). However, the absolute values of the deposition anomaly shown by Kravitz et al. (2009) are larger than in our simulations because their anomaly includes changes in the deposition of tropospheric sulfur (between two simulations with and without SAG), which is not the case in our model. On a global average, wet deposition contributes 68 %, dry deposition 28 %, and sedimentation 4 % to the total deposition rate at the surface.

For estimating the impact in terms of acidic deposition on ecosystems, we convert the deposition rate of sulfur from $\mathrm{mg\,S\,m^{-2}\,yr^{-1}}$ to $\mathrm{mEq\,m^{-2}\,yr^{-1}}$ (used in critical loading studies like Kuylenstierna et al. (2001), cited by Kravitz et al. (2009)) using a conversion factor of $16\,\mathrm{mg\,S} = 1\,\mathrm{mEq}$. With this, the largest additional deposition rates (due to aerosol from SAG) in our simulation are equivalent to $5\,\mathrm{mEq\,m^{-2}\,yr^{-1}}$. This is well below the critical loadings for almost all ecosystems reported by Kuylenstierna et al. (2001) and on the very low side for the most sensitive type of ecosystems (e.g., waterways in Sweden, which have a critical load of $1\text{--}44\,\mathrm{mEq\,m^{-2}\,yr^{-1}}$).

Kravitz et al. (2009) gives a global mean value of $270\,\mathrm{mg\,S\,m^{-2}\,yr^{-1}}$ for the present-day deposition rate of tropospheric sulfur. Compared with this large number, our simulated global mean additional sulfate deposition due to SAG of $18\,\mathrm{mg\,S\,m^{-2}\,yr^{-1}}$, but even the maximum value of $82\,\mathrm{mg\,S\,m^{-2}\,yr^{-1}}$ is rather small.

In order to assess the possible impact on human health, we also compute the concentration of $PM_{2.5}$ at ground level due to aerosol particles coming from the stratosphere. For this we take the sum over the aerosol mass in all size bins with dry diameter lower than $2.5\,\mathrm{\mu m}$, assuming that they consist of ammonium sulfate with a density of $1.77\,\mathrm{g\,cm^{-3}}$. Due to this choice, we probably overestimate $PM_{2.5}$, because the wet radius of some of these particles is actually above $2.5\,\mathrm{\mu m}$. Allowing the particles to grow further in the troposphere would also result in smaller concentrations of small particles.

The additional $PM_{2.5}$ concentration at the surface due to SAG is shown in Fig. 17. It does not exceed $1\,\mathrm{\mu g\,m^{-3}}$ and is largest over the ocean, where the model assumes the lowest dry deposition velocities, thereby increasing the aerosol lifetime in marine boundary layer. In order to get a rough estimation of the impact of this additional $PM_{2.5}$ on human health, we first compute the product of $PM_{2.5}$ from SAG and human population count (2015 values from SSP1, see Riahi et al. (2017)) at the grid cell

level and cumulate these values over the whole Earth. This cumulative sum is $4.2 \cdot 10^9\,\mathrm{\mu g\,m^{-3}}$ person, which is approximately 2% of the value found using present day $PM_{2.5}$ values from van Donkelaar et al. (2016) at $2.4 \cdot 10^{11}\,\mathrm{\mu g\,m^{-3}}$ person. Although previous studies (e.g. Partanen et al., 2013) have translated such changes in surface PM concentrations into mortality or morbidity rates, we do not attempt to do so as the health impact of PM is dependent on aerosol size and chemical composition in ways that are unknown.

## 4    Conclusions

The model results for various scenarios of stratospheric aerosol geoengineering (SAG) presented in this study imply that the net radiative forcing achievable through equatorial SAG with $SO_2$ might be smaller than previously estimated. The radiative heating through the aerosol can disturb the stratospheric dynamics in such a way that the meridional transport is hindered, resulting in larger sulfate concentrations in the tropics, which enhances particle growth. The larger particles are responsible for an important positive LW forcing, which can compensate the negative SW forcing (cooling) almost completely for large $SO_2$ injection rates like $50\,\mathrm{Tg\,S\,yr^{-1}}$ (the maximum rate simulated in this study). We find that it might be impossible to achieve a more negative net instantaneous radiative forcing than $-2\,\mathrm{W\,m^{-2}}$ with equatorial $SO_2$ injections.

We also find that $SO_2$ injections at higher altitude (in the range 17 to 23 km) do not result in larger (i.e., more negative) radiative forcing, because the particles grow to larger size during their stratospheric lifetime and have less optimal optical properties. This finding contradicts previous studies by Niemeier et al. (2011) and Niemeier and Timmreck (2015) but agrees with a recent study by Niemeier and Schmidt (2017).

Enlarging the injection area from one equatorial grid cell to several grid cells between 30°S and 30°N resulted in smaller particles, but also in a smaller global aerosol burden, which in total causes the net radiative forcing to be equal to the one from equatorial injections. Restricting the injections to a shorter period of the year with seasonally varying latitude resulted in a small increase in net radiative forcing, but probably only due to stronger updraft (more heating from larger sulfate concentrations).

The simulated impact on stratospheric dynamics through radiative heating by the aerosol agrees with a previous study by Aquila et al. (2014) in that the QBO breaks down for injection rates larger than about $5\,\mathrm{Tg\,S\,yr^{-1}}$. This results in a stronger subtropical transport barrier, which causes the particles to grow even larger. In contrast, the impact of the additional sulfate at the Earth's surface is probably small. The maximum additional acidic deposition is below the critical loadings for almost all ecosystems and the additional particle concentrations ($PM_{2.5}$) are below $1\,\mathrm{\mu g\,m^{-3}}$, resulting in a small $2\,\%$ increase of population-weighted $PM_{2.5}$ relative to present-day conditions.

In this study we only consider the instantaneous radiative forcing (IRF) by the aerosol. But in the real climate system this IRF will induce rapid adjustments and slower feedback mechanisms. The combination of these ultimately determines the impact of SAG on the climate. For studying the slow feedbacks one would need to couple the atmospheric model at least to an ocean model, which we did not do in the presented experiments. But we studied the rapid adjustments to the IRF, resulting in an

effective radiative forcing (ERF), in a separate paper (Boucher et al., 2017). We found that rapid adjustments significantly enhance ERF relative to IRF because of stratospheric warming and despite a moistening of the stratosphere.

In our experiments we found that the positive LW forcing can compensate a large part of the negative SW forcing, i.e. 67 % in the STANDARD $10\,\mathrm{Tg\,S\,yr^{-1}}$ and 93 % in the $50\,\mathrm{Tg\,S\,yr^{-1}}$ scenario. Therefore, we argue that the LW forcing should always be considered in modelling studies of SAG, which has not often been the case in the past. Considering only the SW forcing of SAG will lead to considerable overestimation of its efficacy.

The simulation results depend largely on the modelled evolution of particle size, radiative heating and stratospheric dynamics (controlling the spatial distribution). The stratospheric processes determining this are complex and uncertain and their implementation causes considerable disagreement between different models. In contrast to volcanic eruptions, models cannot be constrained by observations in the case of purely hypothetical geoengineering scenarios. Therefore, it would be worthwhile to compare the results with those from other models, like we did it in Sec. 3.2. Further intercomparison could increase the robustness of the findings from this study and it could also help to further improve the models by examining differences (e.g. the refractive index and the resulting optical properties of the aerosol) in more detail.

In additional sensitivity experiments including the effect of van der Waals (vdW) forces on coagulation we found that the strong decrease in forcing efficiency for larger injection rates observed in the other experiments (without vdW enhancement) is probably underestimated. But we should note that the exact impact of vdW forces on coagulation is not very well determined from theory and laboratory experiments and that the implementation in our model is only tentative. These results thus justify further work on parametrising aerosol coagulation in models as well as our choice for a sectional approach to represent the aerosol size dynamics.

It has to be noted that the results presented in the previous section are strongly influenced by the choice of the injection scenarios. Spreading the injections over larger areas would lower the local concentrations of precursor gases and sulfate and would therefore probably result in less particle growth and more favourable optical properties. Smaller local extinction by the aerosol would also cause less heating and probably less updraft of the particles, which may increase the importance of the initial injection height. Hence, our finding that higher injections do not result in larger radiative forcing (for a given sulfur mass), may not hold for different injection schemes.

We also have to keep in mind that our model setup does not include chemical feedbacks of the aerosol, e.g. on stratospheric ozone. Induced changes in ozone concentration might alter the radiative effects of SAG (especially the stratospheric heating) considerably. The prescription of sea surface temperatures constitutes another limitation for correctly simulating the feedback of SAG on atmospheric dynamics.

The phenomenon of aerosol lifting through local heating and dynamical changes might also be used for the design of better injection strategies, as it could allow technically less demanding injections at lower altitudes. But as the larger local sulfate concentrations needed for the updraft will probably also enhance particle growth, it is unclear whether such a strategy could be more efficient at all. One could possibly imagine methods to increase buoyancy in the initial aerosol (or aerosol precursor) plume that have less impact on particle growth.

Overall, this study may provide additional evidence that solar radiation management with stratospheric sulfate aerosols (formed from tropical injections of $SO_2$) is still more complicated, probably less effective and may implicate stronger side-effects than initially thought.

*Code and data availability.* The data are available from the authors upon request. The code of S3A can be downloaded along with the
5   LMDZ model from http://lmdz.lmd.jussieu.fr. S3A code is mostly contained within a separate directory StratAer of the model physics and is activated at compilation with a CPP key. A model configuration LMDZORSTRATAER_v6 containing the S3A module is also available within the modipsl/libIGCM model environment of the IPSL Earth System Model http://forge.ipsl.jussieu.fr/igcmg_doc.

*Author contributions.* C. Kleinschmitt developed most of the S3A model, designed and performed the SAG simulations, and analysed the data. He wrote most of the article. O. Boucher and U. Platt assisted in the model development and the analysis of the model simulations.

10   *Competing interests.* The authors declare that they have no competing interests.

*Acknowledgements.* This article is a contribution to the DFG-funded priority program SPP 1689 (project PL 193/15-1) which we gratefully acknowledge. We thank Ulrike Niemeier for sharing detailed results from previous studies by Niemeier and Timmreck (2015) and Niemeier and Schmidt (2017) with us and for her helpful discussion about the different results. The authors acknowledge computing time from the TGCC under the GENCI projects t2016012201 and t2017012201.

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

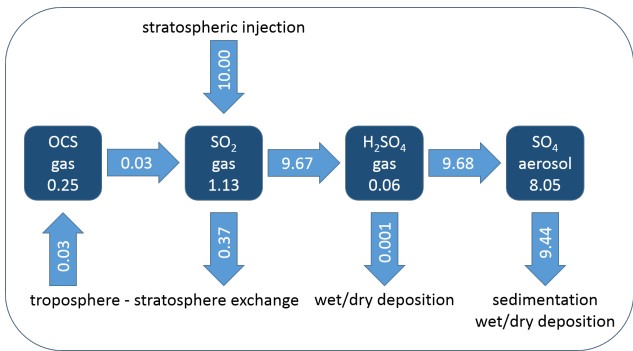

**Figure 1.** Annual mean stratospheric burdens (boxes, in $\mathrm{Tg\,S}$) and fluxes (arrows, in $\mathrm{Tg\,S\,yr^{-1}}$) of the represented sulfur species from the 5$^{th}$ year (quasi-steady state) of the STANDARD experiment. Minor discrepancies in the fluxes indicate that the steady state is not entirely reached after 5 years.

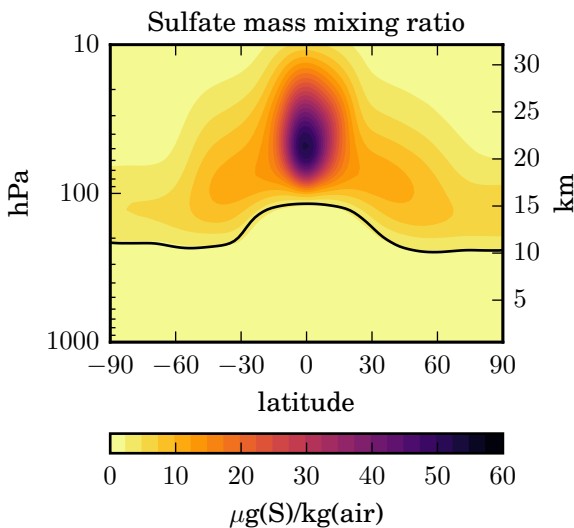

**Figure 2.** Latitude-height cross-section of the zonal and annual mean sulfate aerosol mass mixing ratio (in µg S kg air$^{-1}$) for the STANDARD experiment.

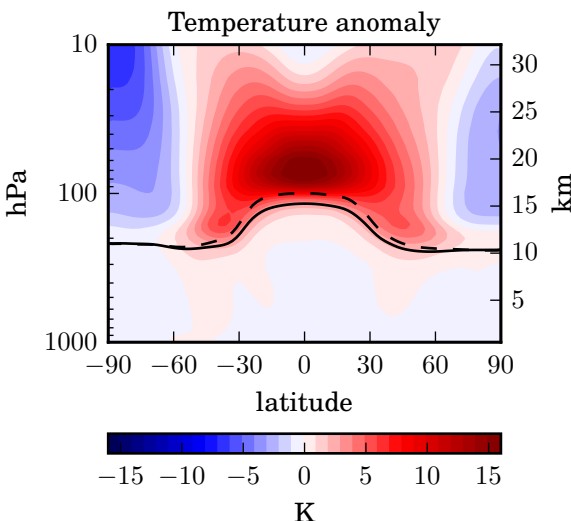

**Figure 3.** Latitude-height cross-section of the zonal and annual mean temperature anomaly (in K) in the STANDARD experiment relative to the CONTROL experiment.

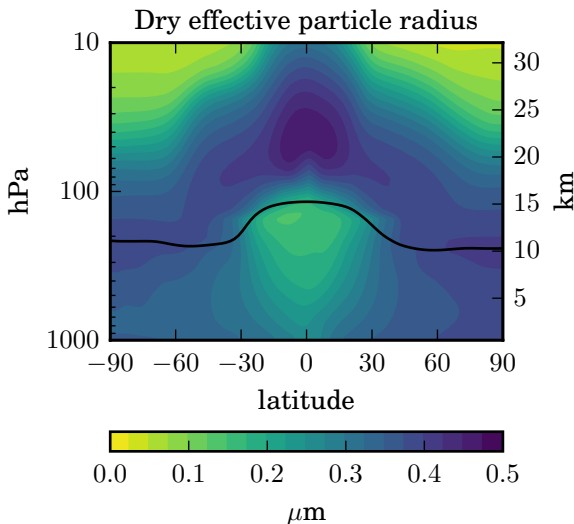

**Figure 4.** Latitude-height cross-section of the zonal and annual mean effective radius (in μm) of dry sulfate particles in the STANDARD experiment. The effective radius is computed from the zonal and annual means of the aerosol volume and surface area.

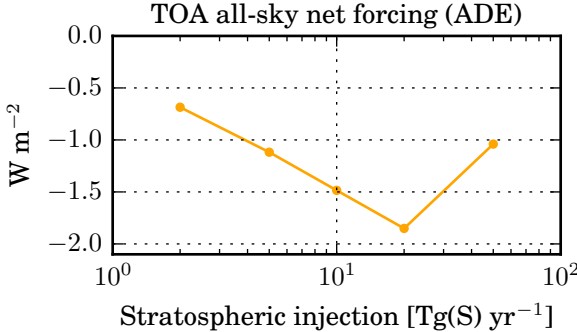

**Figure 5.** Global mean instantaneous net radiative forcing (aerosol direct effect = ADE) at the top of the atmosphere (TOA) in $W\,m^{-2}$ for the $x$_TGS experiments with different injection rates.

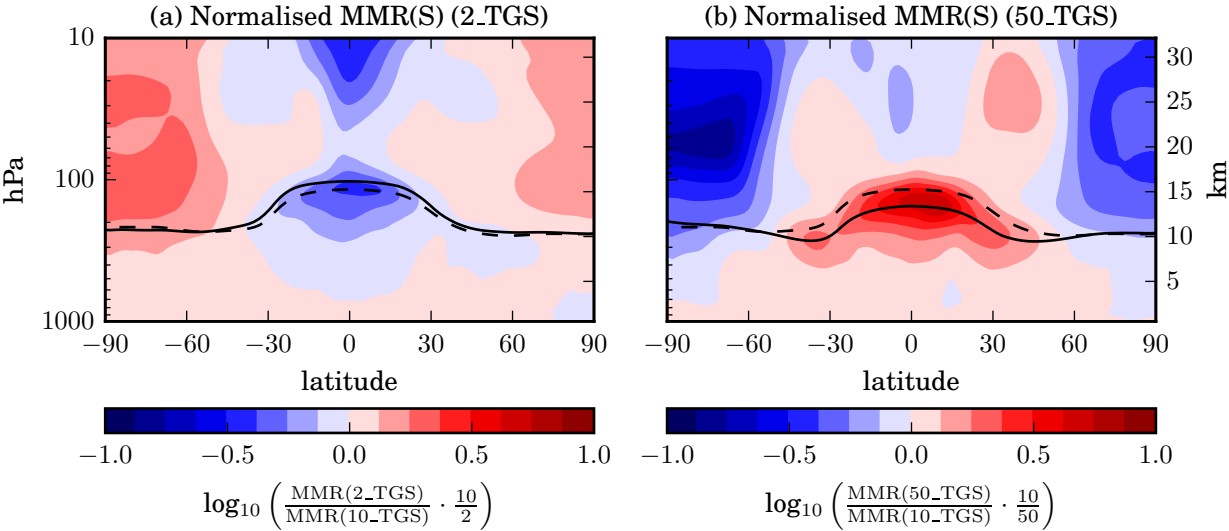

**Figure 6.** Latitude-height cross-sections of the zonal and annual mean sulfate aerosol mass mixing ratio (a) for the 2_TGS and (b) for the 50_TGS experiment, normalised by the value in the STANDARD (= 10_TGS) experiment and the ratio of the injection rates. The base-10 logarithm of the ratio is plotted as indicated below the color scale, so that positive (negative) values indicate larger (smaller) values than implied by a linear scaling. The mean tropopause level in the experiment is indicated by a solid line and the tropopause in the STANDARD simulation by a dashed line.

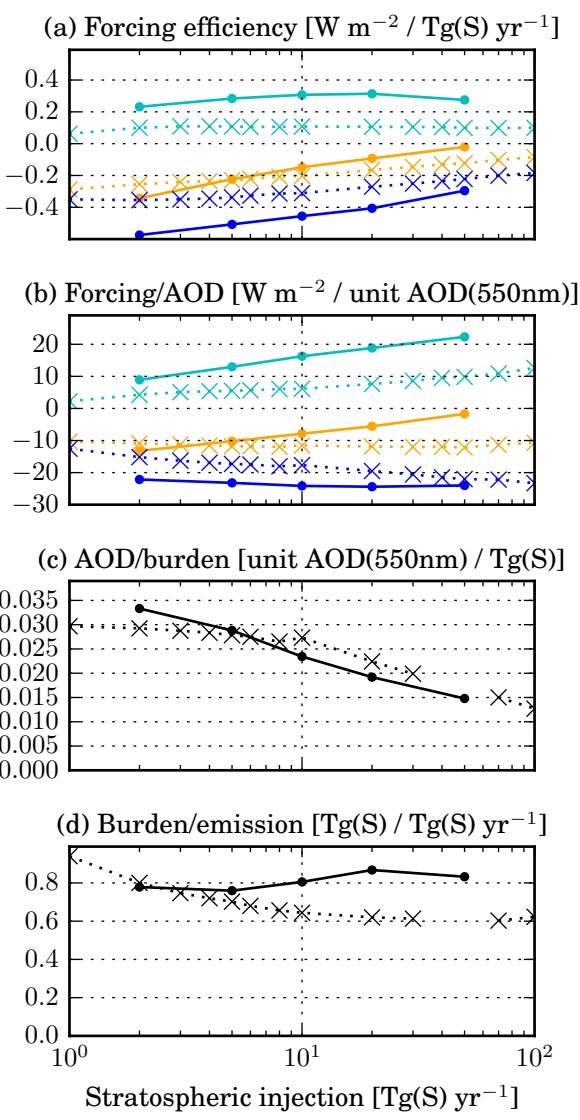

**Figure 7.** (a) Forcing efficiency (i.e., ratio of global mean instantaneous radiative forcing by the global sulfur injection rate) for the $x$_TGS simulations with different injection rates. The SW component is shown in dark blue, the LW component in light blue and the net flux in orange. Panels (b) to (d) show the decomposition of the forcing efficiency as described in the text (see Eq. 1). The solid lines show the results from the described LMDZ-S3A simulations, while the crosses show results from a previous study by Niemeier and Timmreck (2015).

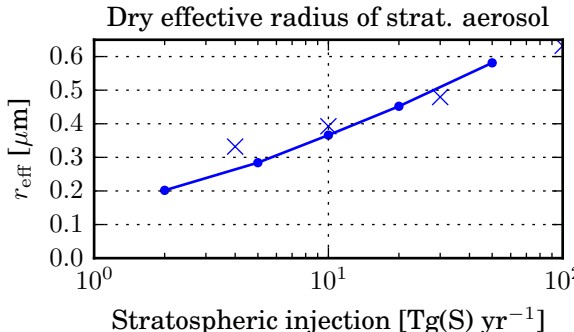

**Figure 8.** Global mean dry effective radius (in μm) of the stratospheric sulfate aerosol as a function of sulfur injection rate (Tg S yr$^{-1}$). The solid lines show the results from the $x$_TGS experiments using LMDZ-S3A, while the crosses show results from a previous study by Niemeier and Timmreck (2015).

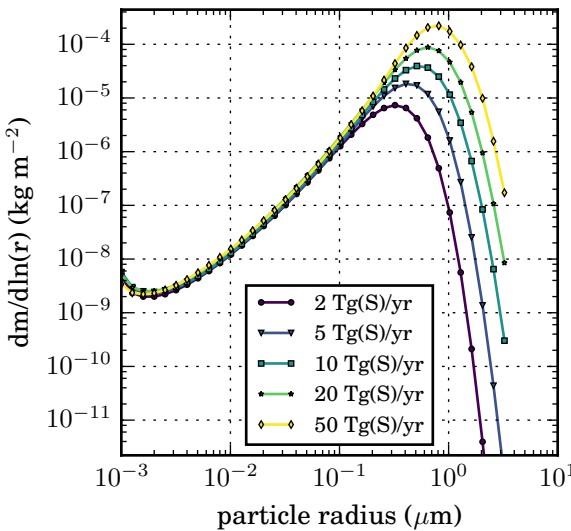

**Figure 9.** Column-integrated global and annual mean stratospheric aerosol mass size distribution ($\mathrm{d}m/\mathrm{d}\ln r$, in $\mathrm{kg(H_2SO_4)\,m^{-2}}$) for the $x$_TGS experiments with varying sulfur injection rate.

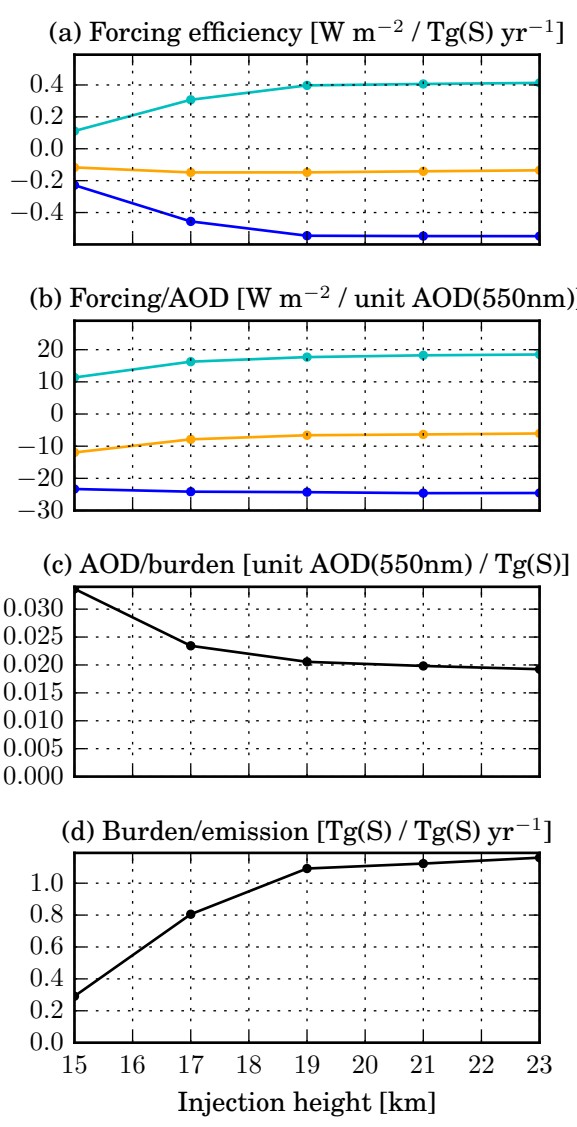

**Figure 10.** Forcing efficiency and its decomposition (as in Fig. 7) for the $z$_KM simulations with different injection heights.

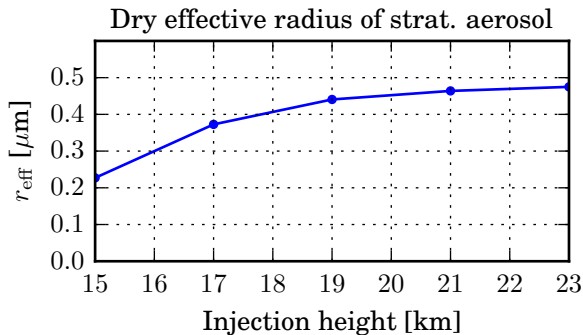

**Figure 11.** Global mean effective radius (µm) of the dry stratospheric sulfate aerosol as a function of injection height (in km) of the $z\_KM$ experiments.

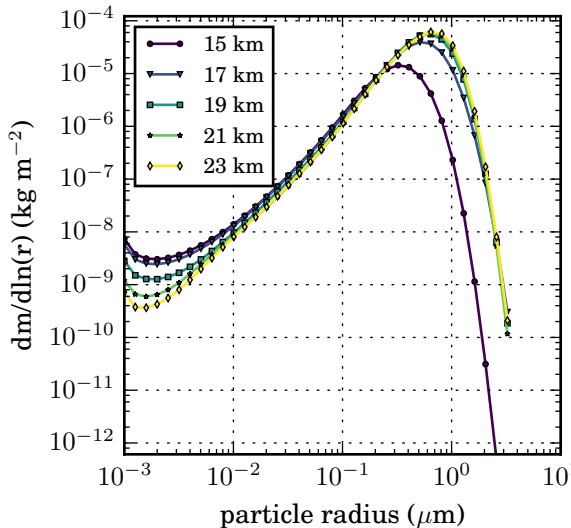

**Figure 12.** Column-integrated global and annual mean stratospheric aerosol mass size distribution ($dm/d \ln r$, in $kg\,m^{-2}$) for the $z\_KM$ experiments with varying injection height.

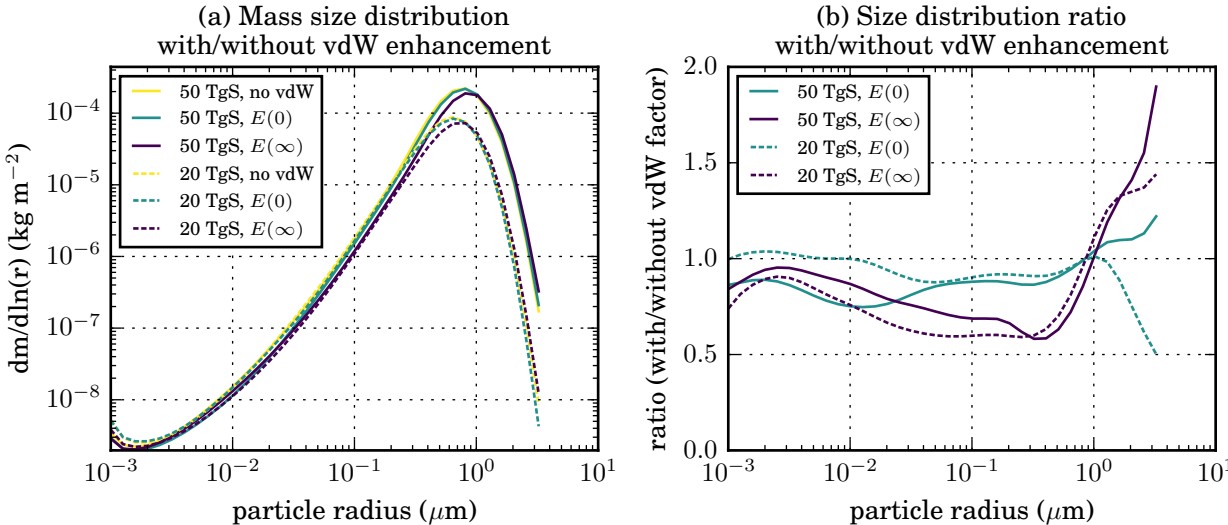

**Figure 13.** (a) Column-integrated global and annual mean stratospheric aerosol mass size distribution ($\mathrm{d}m/\mathrm{d}\ln r$, in $\mathrm{kg\,m^{-2}}$) for the experiments with and without van der Waals (vdW) enhancement of coagulation with the continuum regime enhancement factor $E(0)$ and the kinetic regime enhancement factor $E(\infty)$, respectively. (b) Ratio between the size distributions in the experiments with and without the vdW enhancement factor.

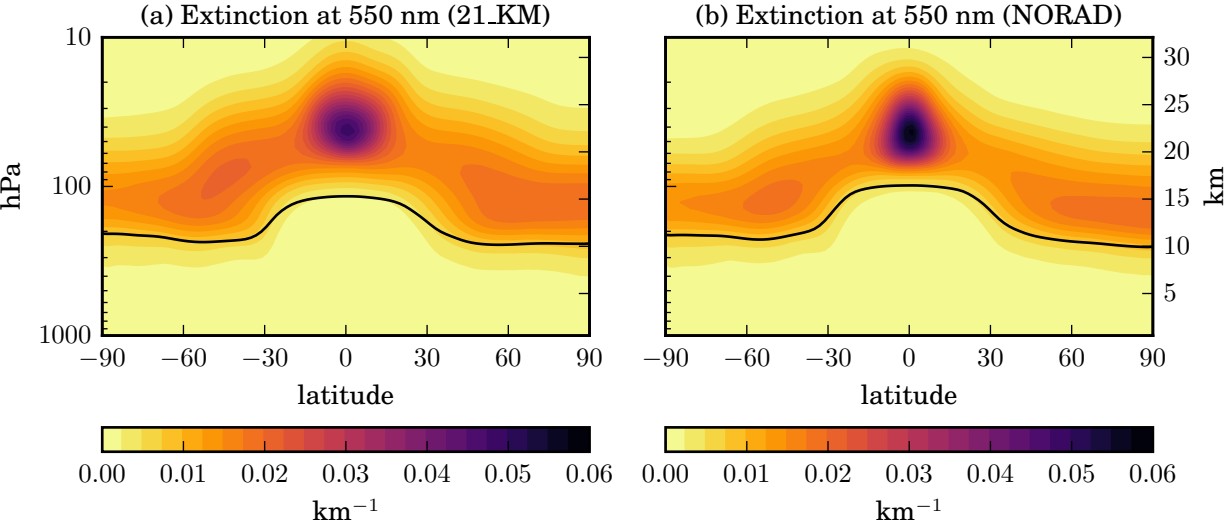

**Figure 14.** Latitude-height cross-section of the zonal and annual mean distribution of the aerosol extinction coefficient (km$^{-1}$) at 550 nm for equatorial stratospheric sulfur injections at 21 km from (a) a simulation with radiatively interactive aerosol (21_KM) and (b) a simulation with non-radiatively interactive aerosol (NORAD).

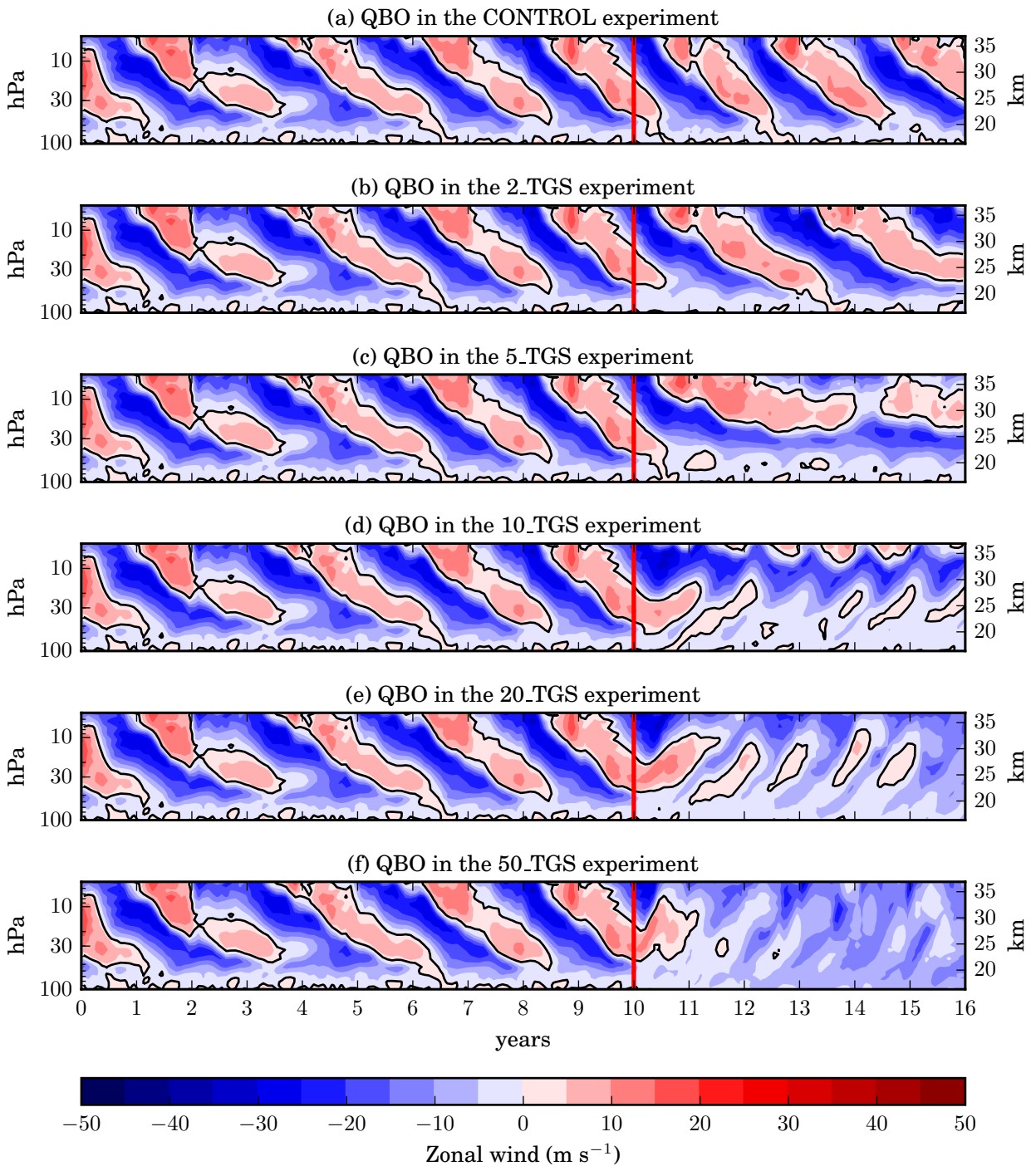

**Figure 15.** Temporal evolution of the vertical profiles of zonal wind, averaged zonally and between 2° N and 2° S. The QBO can be seen in the alternating downward propagation of easterly (blue) and westerly (red) wind direction. All the simulations share the same background aerosol spin up period of 10 years. The onset of the continuous sulfur injections is marked by a red line.

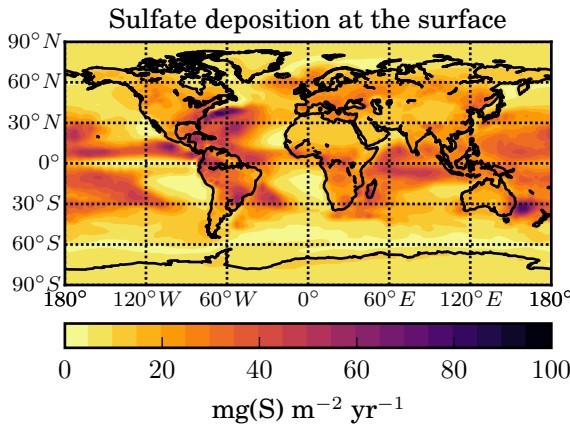

**Figure 16.** Annual mean deposition rate of sulfate aerosol of stratospheric origin at the Earth's surface (in $\mathrm{mg\,S\,m^{-2}\,yr^{-1}}$) in the STANDARD scenario.

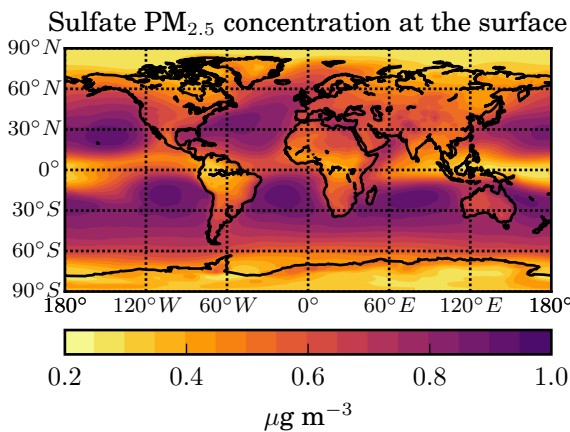

**Figure 17.** Annual mean concentration of sulfate aerosol particles ( as ammonium sulfate) of stratospheric origin with dry diameter below 2.5 μm at the Earth's surface (in μg m$^{-3}$) in the STANDARD scenario.