# Peer review of "Sensitivity of the radiative forcing by stratospheric sulfur geoengineering to the amount and strategy of the SO2 injection studied with the LMDZ-S3A model"

_Atmospheric Chemistry and Physics, 2017_

## Short Comment (SC1) · 4 Sep 2017

I have read your paper with interest and would like to know more about how atmospheric chemistry is treated in your model. This could be important for a variety of reasons. For example, changes in ozone (which would occur under sulfate geoengineering) can affect the calculated radiative forcing, tropopause height, the QBO, the Brewer-Dobson circulation. Since ozone's SW and LW effect are strongly dependent on altitude, this could also affect the small dependency of the calculated RF on injection height (p.9, l.17), in particular if stratospheric temperature adjustments would be

considered. For discussions on some of these effects, see for example:

Pitari, G., V. Aquila, B. Kravitz, A. Robock, S. Watanabe, I. Cionni, N. De Luca, G. Di Genova, E. Mancini, and S. Tilmes (2014), Stratospheric ozone response to sulfate geoengineering: Results from the Geoengineering Model Intercomparison Project (GeoMIP), J. Geophys. Res. Atmos., 119, 2629–2653, doi:10.1002/2013JD020566.

Nowack, P. J., Abraham, N. L., Maycock, A. C., Braesicke, P., Gregory, J. M., Joshi, M. M., Osprey, A., and Pyle, J. A.: A large ozone-circulation feedback and its implications for global warming assessments, Nat. Clim. Change, 5, 41–45, doi:10.1038/nclimate2451, 2015.

Rind, D., J. Jonas, N. K. Balachandran, G. A. Schmidt, and J. Lean (2014), The QBO in two GISS global climate models: 1. Generation of the QBO, J. Geophys. Res. Atmos., 119, 8798–8824, doi:10.1002/2014JD021678.

deF. Forster, P. M., and K. P. Shine (1997), Radiative forcing and temperature trends from stratospheric ozone changes, J. Geophys. Res., 102(D9), 10841–10855, doi:10.1029/96JD03510.

Could you put your results into context depending on how ozone is treated in your model and accordingly add some information to your model description?

In addition, I am highly interested in your point on potential impacts on air quality (PM2.5 etc, section 3.7). Could you discuss how such effects might interact with the treatment of atmospheric chemistry (e.g. OH, ozone fields)? If you have ozone model output, could you compare your surface ozone results briefly with

Nowack, P. J., Abraham, N. L., Braesicke, P., and Pyle, J. A.: Stratospheric ozone changes under solar geoengineering: implications for UV exposure and air quality, Atmos. Chem. Phys., 16, 4191-4203, https://doi.org/10.5194/acp-16-4191-2016, 2016.

Xia, L., Nowack, P. J., Tilmes, S., and Robock, A.: Impacts of Stratospheric Sulfate Geoengineering on Tropospheric Ozone, Atmos. Chem. Phys. Discuss.,

https://doi.org/10.5194/acp-2017-434, accepted for publication.

https://www.atmos-chem-phys-discuss.net/acp-2017-434/

or put your air quality results into a more general context (ozone, NOx etc)?

———————————————————

---

## Referee Comment (RC1) · Anonymous Referee #1 · 5 Sep 2017

This manuscript reports on studies of geoengineering via stratospheric SO2 injection performed with the GCM LMDZ coupled to a sectional aerosol model S3A (the coupled model is called LMDZ-S3A). The model has some limitations (fixed SST, no ozone feedbacks, fixed chemical time scale for SO2 conversion, fixed aerosol composition for radiative calculations) but nevertheless does include important feedbacks (aerosol feedback on radiation, interactive tropopause height, self-generated QBO) for stratospheric aerosol injection (SAI). The paper is generally well-written and the subject matter appropriate to ACP. This paper repeats some experiments of a study performed by Niemeier and Timmreck (2015) but with a different model, and interestingly it reaches somewhat different conclusions. Given the uncertainties in global modeling of something like stratospheric aerosol injection, having model results from multiple independently-formulated models is very desirable. And when models diverge in their conclusions, it may lead us to widen the uncertainty bounds, but also to investigate mechanism, feedbacks, and model reliability.

I support publication of this paper after the following scientific issues are addressed.

Specific Comments:

I would like the paper to emphasize somewhat more that the results presented apply only to injection of SO2 – injection of H2SO4 (see Pierce et al., 2010) or even injection of SO2 outside the tropics, may display different RF responses and scalings. In several places the paper uses "stratospheric sulfate aerosol" or "sulfate SAI" to refer to SO2 injection with statements that may not be as appropriate for injection of H2SO4 or preformed sulfate aerosol particles. The language just needs to be more precise (such as "tropical injection of SO2") to avoid overgeneralizing the conclusions.

The reference Boucher et al., 2017, referred to as a "companion paper" in the introduction, is incomplete. Perhaps this article is currently under review, as searches of GRL do not reveal such a paper. Given that this reference contains analysis of the importance of rapid adjustment of T, H2O and cloud fields to RF, it should have been included with the review materials. It would be useful to show a comparison of radiative forcing generated, not only as the difference in a single simulation calculated with and without aerosols, but also as the SAI scenario minus the CONTROL scenario. This would allow the reader to know the magnitude of the impact of changes in the temperature and H2O on RF. The other part of the RF calculation which should be quantified is the portion of the calculated RF due to background sulfate, which is apparently not subtracted in the current methodology. Calculating RF of the CONTROL case with and without aerosols would provide that.

[Figure]

Text describing Figure 12 is missing. Perhaps an accidental deletion of text on page 10 between "The particle size...appears to approach saturation levels below 0.5 um with increasing injection height." And "Only the aerosol mass on the lower end of the size range decreases further...". The latter part of this paragraph seems to refer to Figure 12.

The "Conclusion" section should mention the lack of chemical feedbacks through ozone.

Technical Corrections:

Page 8, line 23: "monotonously" should be "monotonically".

In Figures 7 and 10, the colors for LW and SW forcing are different shades of blue which are difficult to distinguish. Why not different colors rather than shades of the same color? Also, a legend on the plots describing the different colors and symbols would aid comprehension.

---

## Referee Comment (RC2) · Anonymous Referee #2 · 6 Sep 2017

This paper describes the use of the LMDZ-S3A model to study the efficacy and impacts of stratospheric sulfur geoengineering. This is valuable research, as stratospheric aerosols from many sources (e.g. volcanoes and hypothetical geoengineering schemes) can impact climate and chemistry, and there remains much uncertainty regarding stratospheric processes and aerosol evolution. This is an excellent model for this work, as the model includes aerosol microphysical processes, sectional aerosol size representation, aerosol-radiative interactions in both the SW and LW, high vertical resolution, stratospheric processes including QBO, and precursor gas emissions. The

authors apply this model to study the impacts of equatorial SO2 injections at various magnitudes (2-50 Tg S/yr) and altitudes (15-23km) as well as a seasonal and broad injections, on aerosol properties, radiative fluxes, stratospheric circulation, and surface pollution. The paper is well-written, clearly laid out, and provides valuable contributions to scientific understanding of stratospheric sulfur geoengineering. I have one general/moderate concern and numerous small/specific concerns with the manuscript.

General/moderate concern:

The complexity of aerosol-chemical-dynamic processes, combined with long aerosol lifetime in the stratosphere, results in high uncertainty with models. Indeed, the results of this work differs fairly significantly from some other work, which the authors cite (for example, the large differences in radiative efficacy at higher injection rates compared to Niemeier and Timmreck 2015, and the large differences in efficacy at higher injection altitudes compared to Niemeier et al. 2011, Niemeier and Timmreck 2015). As such, I recommend a few things:

1) Add a paragraph to the introduction, as well as the conclusion, highlighting the significant uncertainties with stratospheric processes, especially hypothetical geoengineering schemes which have no observations to evaluate models against (yet) 2) Add more discussion throughout the paper on possible sources of uncertainties and errors with the current model implementation. A few discussion items: a) Prescribed oxidants: How do you know that prescribed oxidants in your model do not significantly impact results? It would be nice to do a sensitivity study, perhaps varying the availability of oxidants, and calculating how it impacts results. How might its feedback with water vapor impact results? Why did you not quantify ozone destruction, which is a known possible consequence of stratospheric sulfur geoengineering? Discuss. b) Coagulation processes: There are numerous coagulation processes that might be significant for geoengineered stratospheric aerosols – for example, you mention van der Waals forces. As you note, previous studies have found van der Waals forces to impact ambient stratospheric aerosol, and volcanic eruptions. The impacts of van der Waals forces
on geoengineered aerosols seems uncertain, and possibly important. I suggest conducting an experiment with van der Waals forces compared to the standard experiment to see if it impacts results; at a minimum, further discussion. c) Any other limitations in your model worth discussing further?

Specific comments:

Abstract: 1) More clearly state that you calculate the limit of sulfate geoengineering efficacy to be 2 Wm-2, and that efficacy actually decreases at injections larger than 20 Tg S/yr – this is very interesting. 2) Add a caveat that stratospheric processes are complex and uncertain and that multiple modeling studies disagree with one another, including your model.

p1, line 20: suggest that you change the term "physico-chemistry" to something more understandable. Also, does this include impacts of ozone? If so, I suggest clarifying that, e.g. "chemistry and ozone"

p4, line 14: why do you expect the contribution of van der Waals forces to be insignificant? By changing the tails of the aerosol size distributions, van der Waals forces could possibly be significant over a long period of time, even if the impacts were negligable for a Pinatubo-size eruption. (see also my comment at p8, line 25).

p4, line 26: How might prescribed oxidants impact results or introduce error? (do you expect the oxidant concentration to remain constant over a long period of time given continuous so2 injection?)

p6, line 20: Change "completely" to something less exact, perhaps "essentially"

p8, line 25: The difference between your results and that of Niemeier and Timmreck (2015) is interesting. Can you estimate how much of the difference might be due to differences in the particle size distribution? (can you change your particle size distribution and run it through radiation code?) Other studies have found differences in the size distributions between modal models and sectional models. In an intercomparison

study, Weisenstein et al. [2007] found sectional models to predict larger stratospheric aerosols after Pinatubo than modal models, and sectional models compared better to observations. Kokkola et al. [2009] completed an intercomparison between microphysical and sectional versions of the ECHAM model and found significant differences in the evolution of stratospheric aerosol size distributions, with differences between modal and sectional models increasing with increasing $SO_2$ injection rate. English et al [2013] calculated aerosol mode widths from their sectional model assuming a lognormal distribution and found the mode width to change significantly after simulated volcanic eruptions, suggesting that 2-moment modal models with fixed mode widths may not accurately capture the evolving size distribution.

p9, line 32: Please cite Niemeier et al. 2011 who also studied injection height. The disagreement between your results and theirs is also interesting. Niemeier et al. 2011 also predicted increasing particle size with height (Fig 2b in their paper). Why might your results differ from theirs? Differences in LW radiation code? Differences in zonal transport with altitude? QBO?

p10, line 13: Are you injecting only in 28 grid boxes? If so, the concentrations might still be high compared to other studies which continuously inject across all grid boxes spanning a latitude range (e.g. 30 S - 30 N). Also, please clarify the altitude for the Broad injection here.

p11, line 15: Please cite Aquila et al. 2012, who also quantified the impacts that aerosol radiative heating have on the peak injection altitudes (for Mount Pinatubo).

p13, line 22: How does this contradict English et al. (2013)? They did not mention radiative fluxes. Differences in AOD distribution might be because their model does not have aerosol radiative heating coupled.

p13, line 22: Cite Niemeier et al 2011 in addition to Niemeier and Timmreck 2015. Again, why do your results differ if they also found an increase in particle radius with altitude?

p14, line 14: Expand this paragraph to further discuss the difficulties and complexities regarding geoengineered stratospheric aerosol (the processes are complex; the long lifetime allows for errors to amplifiy; geoengineering is only hypothetical and has no observations to evaluate models against (yet); potential uncertainties/errors in your model.

Figure 4: the x-axis units label appears as "m" which is incorrect.

Figure 6: latitude axis shows all "0"

Figure 7d, 8, 10d, 11,12, and 14: y-axis values are cut off

Figure 15: It would also be useful to have a plot of percent increase in deposition rate.

References

Aquila, V., L. D. Oman, R. S. Stolarski, P. R. Colarco, and P. A. Newman (2012), Dispersion of the volcanic sulfate cloud from a Mount Pinatubo–like eruption, J. Geophys. Res., 117, D06216, doi:10.1029/2011JD016968.

Kokkola, H., R. Hommel, J. Kazil, U. Niemeier, A.-I. Partanen, J. Feichter, and C. Timmreck (2009), Aerosol microphysics modules in the framework of the ECHAM5 climate model – intercomparison under stratospheric conditions, Geosci. Model Dev., 2, 97–112, doi:10.5194/gmd-2-97-2009.

Weisenstein, D. K., J. E. Penner, M. Herzog, and X. Liu (2007), Global 2-D intercomparison of sectional and modal aerosol modules, Atmos. Chem. Phys., 7, 2339–2355, doi:10.5194/acp-7-2339-2007.

---

## Referee Comment (RC3) · Anonymous Referee #3 · 6 Sep 2017

This manuscript explore several geoengineering scenarios characterized by different injection amounts, altitudes, latitudes, and time of the year using the LMDZ-S3A model, which includes a sectional aerosol module. Several aspects of the simulations are here analyzed, both regarding the evolution of the aerosol and the effects on stratospheric dynamics.

Despite not being particularly innovative, this is an excellent paper that documents the extensive work performed by the author. It is a through analysis of different geoengineering scenarios within the same model framework, and the results are compared

to previous publications when appropriate. I have only some very minor comments, mostly of technical nature.

1 - Most figures were corrupted when generating the PDF (special characters and the scales of some axis are missing). Please correct them before publication.

2 - Page 6 Line 24: I wonder if also the concentration (ug/m3) of sulfate peaks well above the injection height.

3 - page 7 line 20: "The region with the largest particles descends towards higher latitudes due to ongoing particle growth and sedimentation during the meridional transport through the BDC." I am not sure I agree with this statement. This would imply that the effective radius at higher latitudes is larger than in the tropics, because particles have grown during the transport via BDC. The figure does not show that, though.

4 - page 8 line 7: Figure 7 shows the absolute value of the forcing efficiency, correct?

5 - page 9: Fig. 9 is not referred to in the test, right? I think it is a very useful figure that should be discuss. It can say something about how appropriate (or not) a modal approach is.

6 - page 9 line 18: ", it appears that the net forcing/AOD decreases for higher injections,". Am I looking at the orange line if Fig. 10b? That line increases with injection height.

7 - page 10 line 4: Are the authors referring to Fig. 12?

8 - page 11 line 13: a difference plot would be useful, as the distributions are relatively similar.
* * *

---

## Referee Comment (RC4) · Anonymous Referee #4 · 12 Sep 2017

The paper by Kleinschmitt et al., discusses the sensitivity of radiative forcing by stratospheric aerosol geoengineering to the injection amount and region using the LMDZ-S3A model and a section stratospheric aerosol model. The paper contributes to the question how efficient sulfur geoengineering can counteract radiative forcing. It is therefore an important contribution that should be published. The paper considers injection amounts as large as 50TgS to identify limits of this approach and also explores changes with regard injection locations. All experiments consider injections at the equator or within a region around the equator between 30N to 30S. In contrast to

earlier studies, the amount of radiative forcing with injection amount is limited, which may be a result of the specific setup and the model. The paper is well written and I would recommend the publication to ACP after my comments and suggestions as listed below have been addressed.

General comment: One general comment is about the setup of the model. The use of fixed SSTs and non-interactive chemistry has limitations in correctly simulating the response of sulfur injections to dynamics. This has been mentioned in the manuscript, but a caveat along these lines needs to be also added in the discussion and conclusions of the paper. In addition, various figures in this paper do not show x and y axis annotations correctly.

Specific comments:

Introduction: Line 18: SOA is a misleading abbreviation, aerosols are not injected. Please change to "sulfur injection" or "stratosphere aerosol geoengineering"

Line 22: "using different aerosol types" is not correct. All the listed studies injected SO2 and some used H2SO4.

Page 3, Line 4: Can you give a reference that discusses the changes in the strength of the tropical pipe with regard to the phase of the QBO?

Line 23: What is pressure at the top of the model?

Line 7: Is there a references that supports that commercial aircraft can hardly fly higher then 15km?

Line 10: Is there a reference that shows that injections at several locations may be more expensive and limited economies of scale? Why would injections at multiple locations be more effective? If there are studies discussing this, please mention those.

Page 5, Line 15: The aerosol distribution is expected to adjust after 2 years, which needs to be shown. Even though 5 year simulations are rather short, I would recommend to average the last 3 years of the simulation to include the variability between the years. How will the budget in Figure 1 change if the last 3 years of the simulation are considered?

Page 7, Line 5: Total column ozone has decreased after volcanic eruptions in the past, but ozone mixing ratios do not decrease everywhere in the stratosphere but also increase in the tropics due to the hydrolysis of N2O5 (Fahey et al, 1993).

Page 9: 3.3 Sensitivity to injection height: what is the injection amount used in these sensitivity studies?

Page 13, Line 23: Niemeier and Schmidt, 2017: is this study published in ACP?

Page 14, last sentence: "and may be implicate stronger side-effects than initially thought". What do you mean by this? What are the side effects you referring to?

Figure 2, 3, 4, 6, 13: Latitudes not shown. What years of the simulation are shown?

Figure 7, 8, 9: problems with y-axis

Figure 10, 11, 12: problems with y and x axis, and not clear what injection amount is shown.

Figure 15: how much is the amount compared to the tropospheric burden?

Figure 16: color scale needs to be improved, very difficult to make out any structure.

---

## Author Comment (AC1) · 13 Sep 2017

The referees pointed out to us that a number of figures in the downloadable PDF file of our discussion paper were corrupted in a way that made some of them hard to understand (missing or incomplete labels and units). This was due to a technical problem during the processing of our submitted manuscript. This problem has been resolved and a corrected version of the discussion paper can be downloaded now. We apologise for not having examined the discussion paper in sufficient detail immediately after its publication.

---

## Author Response (AR1)

**Final author response to the Referees' comments on "Sensitivity of the radiative forcing by stratospheric sulfur geoengineering to the amount and strategy of the SO$_2$ injection studied with the LMDZ-S3A model"**

Christoph Kleinschmitt[1,2], Olivier Boucher[3], and Ulrich Platt[1]

[1]Institute of Environmental Physics, Heidelberg University, Im Neuenheimer Feld 229, 69120 Heidelberg, Germany
[2]Laboratoire de Météorologie Dynamique, Institut Pierre-Simon Laplace, CNRS / UPMC / Sorbonne Université, 4 Place Jussieu, 75252 Paris CEDEX 05, France
[3]Institut Pierre-Simon Laplace, CNRS / UPMC / Sorbonne Université, 4 Place Jussieu, 75252 Paris CEDEX 05, France

*Correspondence to:* Christoph Kleinschmitt (christoph.kleinschmitt@iup.uni-heidelberg.de)

We thank the four Referees for their thorough evaluation of the manuscript. Their comments are repeated below in blue and our response follows in black.

**Response to comments by Referee 1**

This manuscript reports on studies of geoengineering via stratospheric SO2 injection performed with the GCM LMDZ coupled
5    to a sectional aerosol model S3A (the coupled model is called LMDZ-S3A). The model has some limitations (fixed SST, no ozone feedbacks, fixed chemical time scale for SO2 conversion, fixed aerosol composition for radiative calculations) but nevertheless does include important feedbacks (aerosol feedback on radiation, interactive tropopause height, self-generated QBO) for stratospheric aerosol injection (SAI). The paper is generally well-written and the subject matter appropriate to ACP. This paper repeats some experiments of a study performed by Niemeier and Timmreck (2015) but with a different model, and inter-
10    estingly it reaches somewhat different conclusions. Given the uncertainties in global modeling of something like stratospheric aerosol injection, having model results from multiple independently-formulated models is very desirable. And when models diverge in their conclusions, it may lead us to widen the uncertainty bounds, but also to investigate mechanism, feedbacks, and model reliability.

I support publication of this paper after the following scientific issues are addressed.

15    We thank the referee for his/her appreciation and the accurate summary of our study.

Specific Comments:

I would like the paper to emphasize somewhat more that the results presented apply only to injection of SO2 – injection of H2SO4 (see Pierce et al. (2010)) or even injection of SO2 outside the tropics, may display different RF responses and scalings. In several places the paper uses "stratospheric sulfate aerosol" or "sulfate SAI" to refer to SO2 injection with statements that
20    may not be as appropriate for injection of H2SO4 or preformed sulfate aerosol particles. The language just needs to be more precise (such as "tropical injection of SO2") to avoid overgeneralizing the conclusions.

We agree with the reviewer and have corrected the manuscript to be more precise on the injection techniques tested in the study.

The reference Boucher et al. (2017), referred to as a "companion paper" in the introduction, is incomplete. Perhaps this article is currently under review, as searches of GRL do not reveal such a paper. Given that this reference contains analysis of the importance of rapid adjustment of T, H2O and cloud fields to RF, it should have been included with the review materials.

We apologize to the reviewer if this has hindered his/her review in some way. We would have been happy to communicate the submitted paper to the reviewer via the ACP editorial office. In any case this "companion" paper is now published under doi:10.1002/2017GL074647.

It would be useful to show a comparison of radiative forcing generated, not only as the difference in a single simulation calculated with and without aerosols, but also as the SAI scenario minus the CONTROL scenario. This would allow the reader to know the magnitude of the impact of changes in the temperature and H2O on RF. The other part of the RF calculation which should be quantified is the portion of the calculated RF due to background sulfate, which is apparently not subtracted in the current methodology. Calculating RF of the CONTROL case with and without aerosols would provide that.

The background sulfate aerosol has a global and annual mean net instantaneous radiative forcing of $-0.04\,\mathrm{W\,m^{-2}}$ ($-0.05\,\mathrm{W\,m^{-2}}$ in the SW and $+0.01\,\mathrm{W\,m^{-2}}$ in the LW). We added this information to the text, but do not subtract it from the presented RF values of SAI, since it may be considered as negligible.

Text describing Figure 12 is missing. Perhaps an accidental deletion of text on page 10 between "The particle size. . .appears to approach saturation levels below 0.5 um with increasing injection height." And "Only the aerosol mass on the lower end of the size range decreases further. . .". The latter part of this paragraph seems to refer to Figure 12.

Thank you for pointing this issue. We added a reference to Figure 12 to this sentence.

The "Conclusion" section should mention the lack of chemical feedbacks through ozone.

We added a sentence on chemical feedbacks and ozone to the conclusions.

Technical Corrections:

Page 8, line 23: "monotonously" should be "monotonically".

Done.

In Figures 7 and 10, the colors for LW and SW forcing are different shades of blue which are difficult to distinguish. Why not different colors rather than shades of the same color? Also, a legend on the plots describing the different colors and symbols would aid comprehension.

The choice of colours is always difficult and is often a matter of personal preferences. We chose these colours as they were already used by Niemeier and Timmreck (2015) to illustrate SW, LW and net radiative forcing.

**Response to comments by Referee 2**

This paper describes the use of the LMDZ-S3A model to study the efficacy and impacts of stratospheric sulfur geoengineering. This is valuable research, as stratospheric aerosols from many sources (e.g. volcanoes and hypothetical geoengineering schemes) can impact climate and chemistry, and there remains much uncertainty regarding stratospheric processes and aerosol

evolution. This is an excellent model for this work, as the model includes aerosol microphysical processes, sectional aerosol size representation, aerosol-radiative interactions in both the SW and LW, high vertical resolution, stratospheric processes including QBO, and precursor gas emissions. The authors apply this model to study the impacts of equatorial SO2 injections at various magnitudes (2-50 Tg S/yr) and altitudes (15-23km) as well as a seasonal and broad injections, on aerosol properties,

5    radiative fluxes, stratospheric circulation, and surface pollution. The paper is well-written, clearly laid out, and provides valuable contributions to scientific understanding of stratospheric sulfur geoengineering. I have one general/moderate concern and numerous small/specific concerns with the manuscript.

We thank the reviewer for his/her positive comments.

General/moderate concern:

10    The complexity of aerosol-chemical-dynamic processes, combined with long aerosol lifetime in the stratosphere, results in high uncertainty with models. Indeed, the results of this work differs fairly significantly from some other work, which the authors cite (for example, the large differences in radiative efficacy at higher injection rates compared to Niemeier and Timmreck (2015), and the large differences in efficacy at higher injection altitudes compared to Niemeier et al. (2011); Niemeier and Timmreck (2015)). As such, I recommend a few things:

15    1) Add a paragraph to the introduction, as well as the conclusion, highlighting the significant uncertainties with stratospheric processes, especially hypothetical geoengineering schemes which have no observations to evaluate models against (yet)

We added the following sentence: "The stratospheric processes determining this are complex and uncertain and their implementation causes considerable disagreement between different models. In contrast to volcanic eruptions, models cannot be constrained by observations in the case of purely hypothetical geoengineering scenarios."

20    2) Add more discussion throughout the paper on possible sources of uncertainties and errors with the current model implementation.

A few discussion items:

a) Prescribed oxidants: How do you know that prescribed oxidants in your model do not significantly impact results? It would be nice to do a sensitivity study, perhaps varying the availability of oxidants, and calculating how it impacts results. How

25    might its feedback with water vapor impact results? Why did you not quantify ozone destruction, which is a known possible consequence of stratospheric sulfur geoengineering? Discuss.

We performed a sensitivity study for the prescribed chemical lifetime of $SO_2$ under Pinatubo conditions in Kleinschmitt et al. (2017) (see Fig. 14 therein). A doubling of the lifetime on the day of the eruption, decreasing linearly to normal values within 1 month, had a relatively small impact on the resulting AOD by delaying and increasing the peak AOD slightly. The impact

30    can be expected to be even smaller for scenarios of continuous SAI with moderate $SO_2$ injection rates, as they result in lower local $SO_2$ concentrations than a sudden major volcanic injection of $SO_2$.

The impact of SAI on stratospheric water vapour is discussed in Boucher et al. (2017). The greenhouse effect of the additional water vapour (1 to 3 ppmv in the STANDARD experiment) partly compensates the negative radiative forcing of the aerosol. As a feedback on stratospheric chemistry is not included in our simulations, we cannot quantify ozone destruction.

35    We added the following sentence to Sect. 2.1 discussing the issue of prescribed oxidants: "For continuous SAI schemes, where

the aerosol layer and its properties reach an equilibrium after several years, we would expect oxidants to reach equilibrium concentrations (somewhat) lower than the climatological values. Assuming a climatological chemical lifetime of $SO_2$ could therefore cause the model to overestimate $SO_2$ to $H_2SO_4$ conversion rates to a certain degree, favouring new particle formation over condensational growth."

 b) Coagulation processes: There are numerous coagulation processes that might be significant for geoengineered stratospheric aerosols – for example, you mention van der Waals forces. As you note, previous studies have found van der Waals forces to impact ambient stratospheric aerosol, and volcanic eruptions. The impacts of van der Waals forces on geoengineered aerosols seems uncertain, and possibly important. I suggest conducting an experiment with van der Waals forces compared to the standard experiment to see if it impacts results; at a minimum, further discussion.

10 In Kleinschmitt et al. (2017) we also performed a sensitivity study for the impact of van der Waals forces under Pinatubo conditions (see Figs. 12 and 13 therein). We found that including them slightly improved the agreement of the modelled particle number and size with observations, but deteriorated the agreement in AOD. Given these mixed results and the uncertainty of the van der Waals coagulation term, we chose not to include this effect in the SAI experiments.

However, for the avoidance of doubt we performed additional experiments where the van der Waals (vdW) effect on coag-
15 ulation is included. The results show that vdW enhancement may further increase particle size and considerably reduce the radiative forcing for larger injection rates. This is now discussed in the manuscript in the new Section 3.5 ("Sensitivity to van der Waals coagulation enhancement factor").

c) Any other limitations in your model worth discussing further?

The use of fixed sea surface temperatures (SSTs) might be another limitation (see response to Referee 4). Therefore we added
20 the following sentence to the conclusions: "The prescription of sea surface temperatures constitutes another limitation for correctly simulating the feedback of SAI on atmospheric dynamics."

Specific comments:

Abstract:

1) More clearly state that you calculate the limit of sulfate geoengineering efficacy to be 2 Wm-2, and that efficacy actually
25 decreases at injections larger than 20 Tg S/yr – this is very interesting.

We modified the sentence in the following way: "As a result, the net instantaneous radiative forcing does not exceed the limit of $-2\,\mathrm{W\,m^{-2}}$ for continuous equatorial $SO_2$ injections and it decreases (in absolute value) for injection rates larger than $20\,\mathrm{Tg\,S\,yr^{-1}}$."

2) Add a caveat that stratospheric processes are complex and uncertain and that multiple modeling studies disagree with one
30 another, including your model.

We added the following sentences: "But it has to be noted that the complexity and uncertainty of stratospheric processes cause considerable disagreement between different modelling studies of stratospheric aerosol geoengineering. This may be addressed through detailed model intercomparison activities, as observations to constrain the simulations of stratospheric aerosol injection are not available and analogues (such as volcanic eruptions) are imperfect."

35 p1, line 20: suggest that you change the term "physico-chemistry" to something more understandable. Also, does this include

impacts of ozone? If so, I suggest clarifying that, e.g. "chemistry and ozone".

We modified the sentence in the following way: "understanding the physics and chemistry of sulfate stratospheric aerosols, their radiative impacts and other potential impacts on the stratosphere, e.g. on ozone".

p4, line 14: why do you expect the contribution of van der Waals forces to be insignificant? By changing the tails of the aerosol size distributions, van der Waals forces could possibly be significant over a long period of time, even if the impacts were negligible for a Pinatubo-size eruption. (see also my comment at p8, line 25).

As discussed above we have performed additional experiments including vdW enhancement and discuss them in the revised manuscript.

p4, line 26: How might prescribed oxidants impact results or introduce error? (do you expect the oxidant concentration to remain constant over a long period of time given continuous so2 injection?)

We expanded the paragraph by the following sentences: "For continuous SAI schemes, where the aerosol layer and its properties reach an equilibrium after several years, we would expect oxidants to reach equilibrium concentrations (somewhat) lower than the climatological values. Assuming a climatological chemical lifetime of $SO_2$ could therefore cause the model to overestimate $SO_2$ to $H_2SO_4$ conversion rates to a certain degree, favouring new particle formation over condensational growth."

p6, line 20: Change "completely" to something less exact, perhaps "essentially"

Done.

p8, line 25: The difference between your results and that of Niemeier and Timmreck (2015) is interesting. Can you estimate how much of the difference might be due to differences in the particle size distribution? (can you change your particle size distri- bution and run it through radiation code?) Other studies have found differences in the size distributions between modal models and sectional models. In an intercomparison study, Weisenstein et al. (2007) found sectional models to predict larger stratospheric aerosols after Pinatubo than modal models, and sectional models compared better to observations. Kokkola et al. (2009) completed an intercomparison between micro- physical and sectional versions of the ECHAM model and found significant differences in the evolution of stratospheric aerosol size distributions, with differences between modal and sectional models increasing with increasing SO2 injection rate. English et al. (2013) calculated aerosol mode widths from their sectional model assuming a log- normal distribution and found the mode width to change significantly after simulated volcanic eruptions, suggesting that 2-moment modal models with fixed mode widths may not accurately capture the evolving size distribution.

The are various differences between our model setup and that of Niemeier and Timmreck (2015) and it would be cumbersome to attribute the difference in results to the different factors. Going beyond the short analysis presented in the manuscript would require a dedicated intercomparison project such as GeoMIP. Differences in size distribution may indeed be a source of differences in the SW and LW RF but it is not appropriate to speculate based on global mean differences alone as geographical variation and details of the size distribution may matter. Differences in refractive indices may also explain some of the differences.

p9, line 32: Please cite Niemeier et al. (2011) who also studied injection height. The disagreement between your results and theirs is also interesting. Niemeier et al. (2011) also predicted increasing particle size with height (Fig 2b in their paper). Why might your results differ from theirs? Differences in LW radiation code? Differences in zonal transport with altitude? QBO?

Again answering these questions would require a significant amount of work and a dedicated intercomparison project with an agreed protocol. The model differences highlighted in our study may convince other modelling groups that such an intercomparison exercise is required in the future.

p10, line 13: Are you injecting only in 28 grid boxes? If so, the concentrations might still be high compared to other studies which continuously inject across all grid boxes spanning a latitude range (e.g. 30 S - 30 N). Also, please clarify the altitude for the Broad injection here.

Yes, in the BROAD scenario the injection take place in only 28 grid boxes. The injection height is the same as in the STANDARD scenario, i.e. 17±1 km everywhere.

p11, line 15: Please cite Aquila et al. (2012), who also quantified the impacts that aerosol radiative heating have on the peak injection altitudes (for Mount Pinatubo).

Done.

p13, line 22: How does this contradict English et al. (2013)? They did not mention radiative fluxes. Differences in AOD distribution might be because their model does not have aerosol radiative heating coupled.

Actually, we wanted to refer to English et al. (2012) instead of English et al. (2013). It is correct that English et al. (2012) only discuss the increase in aerosol burden and AOD for a higher injection altitude, not the resulting radiative forcing. We therefore removed the citation in this place.

p13, line 22: Cite Niemeier et al. (2011) in addition to Niemeier and Timmreck (2015). Again, why do your results differ if they also found an increase in particle radius with altitude?

We added a reference to Niemeier et al. (2011). It remains unclear why exactly they found an increase in radiative forcing for higher injection altitude despite the increase in particle size, while in our case the radiative forcing does not increase for higher injection altitudes.

p14, line 14: Expand this paragraph to further discuss the difficulties and complexities regarding geoengineered stratospheric aerosol (the processes are complex; the long lifetime allows for errors to amplifiy; geoengineering is only hypothetical and has no observations to evaluate models against (yet); potential uncertainties/errors in your model.

We modified the text in the following way: "The simulation results depend largely on the modelled evolution of particle size, radiative heating and stratospheric dynamics (controlling the spatial distribution). The stratospheric processes determining this are complex and uncertain and their implementation causes considerable disagreement between different models. In contrast to volcanic eruptions, models cannot be constrained by observations in the case of purely hypothetical geoengineering scenarios. Therefore, it would be worthwhile [...]".

Figure 4: the x-axis units label appears as "m" which is incorrect.

Figure 6: latitude axis shows all "0"

Figure 7d, 8, 10d, 11,12, and 14: y-axis values are cut off

We apologize for this. The figures got corrupted during the production process of the discussion paper and have already been corrected.

Figure 15: It would also be useful to have a plot of percent increase in deposition rate.

We agree that this would be an interesting information. But as our model (in the configuration used in this study with the S3A module) does not include deposition fluxes of tropospheric sulfate, it is not possible to compare the deposition of SAI sulfate to the deposition of tropospheric sulfate in detail.

**5 Response to comments by Referee 3**

This manuscript explore several geoengineering scenarios characterized by different injection amounts, altitudes, latitudes, and time of the year using the LMDZ-S3A model, which includes a sectional aerosol module. Several aspects of the simulations are here analyzed, both regarding the evolution of the aerosol and the effects on stratospheric dynamics.
Despite not being particularly innovative, this is an excellent paper that documents the extensive work performed by the author. It is a through analysis of different geoengineering scenarios within the same model framework, and the results are compared to previous publications when appropriate. I have only some very minor comments, mostly of technical nature.
We thank the reviewer for his/her general appreciation.

1 - Most figures were corrupted when generating the PDF (special characters and the scales of some axis are missing). Please correct them before publication.
We apologize for this. The figures got corrupted during the production process of the discussion paper and have already been corrected.

2 - Page 6 Line 24: I wonder if also the concentration (ug/m3) of sulfate peaks well above the injection height.
Figure 1 in this reply shows that the sulfate aerosol concentration in $\mu g \, S \, m^{-3}$ peaks at about $19 \, km$, i.e. above the injection height of $17 \, km$ but below the peak of the sulfate mass mixing ratio. We added this information to the text, but do not show the concentration plot in the revised manuscript.

3 - page 7 line 20: "The region with the largest particles descends towards higher latitudes due to ongoing particle growth and sedimentation during the meridional transport through the BDC." I am not sure I agree with this statement. This would imply that the effective radius at higher latitudes is larger than in the tropics, because particles have grown during the transport via BDC. The figure does not show that, though.
This is correct. We removed the words "due to ongoing particle growth" from this sentence.

4 - page 8 line 7: Figure 7 shows the absolute value of the forcing efficiency, correct?
No, the forcing efficiency is shown with positive and negative values, but the labels were corrupted in the previous version.

5 - page 9: Fig. 9 is not referred to in the test, right? I think it is a very useful figure that should be discuss. It can say something about how appropriate (or not) a modal approach is.
We added a reference to Figure 9 in the text.

6 - page 9 line 18: ", it appears that the net forcing/AOD decreases for higher injec- tions,". Am I looking at the orange line if Fig. 10b? That line increases with injection height.
This misunderstanding may have been caused by the corrupted axis labels. The negative value of the net forcing/AOD is approaching zero, i.e. its absolute value decreases with injection height.

[Figure]

**Figure 1.** Latitude-height cross-section of the zonal and annual mean sulfate aerosol concentration (in $\mu g\,S\,kg\,m^{-3}$) for the STANDARD experiment.

Yes, a reference to Figure 12 has been added.
8 - page 11 line 13: a difference plot would be useful, as the distributions are relatively similar.
Figure 2 in this reply shows the difference in aerosol extinction between the experiments. However, we decided to keep the
5    absolute extinction plot in the manuscript.

**Response to comments by Referee 4**

The paper by Kleinschmitt et al., discusses the sensitivity of radiative forcing by stratospheric aerosol geoengineering to the
injection amount and region using the LMDZ-S3A model and a section stratospheric aerosol model. The paper contributes
to the question how efficient sulfur geoengineering can counteract radiative forcing. It is therefore an important contribution
10    that should be published. The paper considers injection amounts as large as 50TgS to identify limits of this approach and also
explores changes with regard injection locations. All experiments consider injections at the equator or within a region around
the equator between 30N to 30S. In contrast to earlier studies, the amount of radiative forcing with injection amount is limited,
which may be a result of the specific setup and the model. The paper is well written and I would recommend the publication to
ACP after my comments and suggestions as listed below have been addressed.
15    We thank the reviewer for his/her positive comments.
General comment: One general comment is about the setup of the model. The use of fixed SSTs and non-interactive chemistry
has limitations in correctly simulating the response of sulfur injections to dynamics. This has been mentioned in the manuscript,

[Figure]

**Figure 2.** Latitude-height cross-section of the difference in the zonal and annual mean distribution of the aerosol extinction coefficient $(km^{-1})$ at 550 nm for equatorial stratospheric aerosol injections at 21 km between a simulation with non-radiatively interactive aerosol (NORAD) and a simulation with radiatively interactive aerosol (21_KM).

but a caveat along these lines needs to be also added in the discussion and conclusions of the paper.

We added the following paragraph to the conclusions: "We also have to keep in mind that our model setup does not include chemical feedbacks of the aerosol, e.g. on stratospheric ozone. Induced changes in ozone concentration might alter the radiative effects of SAI (especially the stratospheric heating) considerably. The prescription of sea surface temperatures constitutes

5 another limitation for correctly simulating the feedback of SAI on atmospheric dynamics."

In addition, various figures in this paper do not show x and y axis annotations correctly.

We apologize for this. The figures got corrupted during the production process of the discussion paper and have already been corrected.

Specific comments:

10 Introduction: Line 18: SAI is a misleading abbreviation, aerosols are not injected. Please change to "sulfur injection" or "stratosphere aerosol geoengineering"

We agree that the term is not very precise, but we chose to keep it anyhow as it is widely used for the injection of aerosol precursors. We added the following sentence to the introduction: "Although we inject an aerosol precursor (i.e. $SO_2$) rather than sulfate aerosol, we loosely refer to the widely used term stratospheric aerosol injection (SAI) in this study."

15 Line 22: "using different aerosol types" is not correct. All the listed studies injected SO2 and some used H2SO4.

Ferraro et al. (2011) did study other aerosol types (titania, limestone, and soot). We believe the sentence is correct as it stands.

Page 3, Line 4: Can you give a reference that discusses the changes in the strength of the tropical pipe with regard to the phase

of the QBO?

We added a reference to Punge et al. (2009).

Line 23: What is pressure at the top of the model?

The top of the model is at a pressure of 4 Pa (i.e. about 80 km altitude). We added this information to the text.

Line 7: Is there a references that supports that commercial aircraft can hardly fly higher then 15km?

McClellan et al. (2012) state that "existing aircraft [...] are not optimized for high altitude flight and therefore are poorly suited to the geoengineering mission". But 15 km are not a hard limit, so we removed it from this sentence.

Line 10: Is there a reference that shows that injections at several locations may be more expensive and limited economies of scale? Why would injections at multiple locations be more effective? If there are studies discussing this, please mention those.

Of course this is somewhat speculative. We have changed the manuscript to say "There are few studies that look at the economic cost of SAI. However, we may naïvely expect that [...]"

Page 5, Line 15: The aerosol distribution is expected to adjust after 2 years, which needs to be shown. Even though 5 year simulations are rather short, I would recommend to average the last 3 years of the simulation to include the variability between the years. How will the budget in Figure 1 change if the last 3 years of the simulation are considered?

Figure 3 in this reply shows that the aerosol burden and size stabilise after 3 to 4 years and that the year to year variability is small. Therefore, we consider the results of the 5th year as representative. We added this information to the text.

[Figure]

**Figure 3.** Time evolution of the global mean stratospheric sulfur burden (left, in Tg S) and sulfate aerosol effective radius (right, in μm) in the STANDARD experiment.

Page 7, Line 5: Total column ozone has decreased after volcanic eruptions in the past, but ozone mixing ratios do not decrease

everywhere in the stratosphere but also increase in the tropics due to the hydrolysis of N2O5 (Fahey et al., 1993).

We modified the sentence in the following way: "[...] whereas $O_3$ mixing ratios are known to have decreased in large parts of the stratosphere after Pinatubo, thus decreasing absorption of SW radiation. However, $O_3$ mixing ratios may also increase in the tropics due to the hydrolysis of $N_2O_5$ (Fahey et al., 1993), so the exact ozone feedback is unclear."

Page 9: 3.3 Sensitivity to injection height: what is the injection amount used in these sensitivity studies?

The injected amount in the experiments with different injection height is always the same as in the STANDARD experiment, i.e. $10\,\mathrm{Tg\,S\,yr^{-1}}$. We added this information to the first sentence of Sect. 3.3.

Page 13, Line 23: Niemeier and Schmidt (2017): is this study published in ACP?

The discussion paper has been online in ACPD since May 2017 and in the meantime the study has been accepted for publication in ACP. The reference was updated in the revised manuscript.

Page 14, last sentence: "and may be implicate stronger side-effects than initially thought". What do you mean by this? What are the side effects you referring to?

We refer to the strong stratospheric heating and to its effects, i.e. the breakdown of the QBO and the rapid adjustments discussed in more detail in Boucher et al. (2017).

Figure 2, 3, 4, 6, 13: Latitudes not shown. What years of the simulation are shown?

The latitudes are shown in the corrected figures. As described in the text, all figures show results from the 5th year of the simulations, if not specified differently.

Figure 7, 8, 9: problems with y-axis

Done.

Figure 10, 11, 12: problems with y and x axis, and not clear what injection amount is shown.

Done.

Figure 15: how much is the amount compared to the tropospheric burden?

The numbers are compared in the text (page 12, line 4 and line 28).

Figure 16: color scale needs to be improved, very difficult to make out any structure.

We have improved the colour scale, but there is actually only little structure to see.

**Summary of changes made in the revised manuscript**

The changes in the manuscript resulting from the comments by the Referees are marked on the following pages.

[revised manuscript text omitted]

---

## Author Response (AR2)

**Author response to further referee comments on "Sensitivity of the radiative forcing by stratospheric sulfur geoengineering to the amount and strategy of the SO$_2$ injection studied with the LMDZ-S3A model"**

Christoph Kleinschmitt[1,2], Olivier Boucher[3], and Ulrich Platt[1]

[1]Institute of Environmental Physics, Heidelberg University, Im Neuenheimer Feld 229, 69120 Heidelberg, Germany
[2]Laboratoire de Météorologie Dynamique, Institut Pierre-Simon Laplace, CNRS / UPMC / Sorbonne Université, 4 Place Jussieu, 75252 Paris CEDEX 05, France
[3]Institut Pierre-Simon Laplace, CNRS / UPMC / Sorbonne Université, 4 Place Jussieu, 75252 Paris Cedex 05, France

*Correspondence to:* Christoph Kleinschmitt (christoph.kleinschmitt@iup.uni-heidelberg.de)

The Referee's comments are repeated below in blue and our response follows in black.

The new version of the paper has improved compared to the previous one. However, a couple of my earlier comments were not sufficiently addressed and are listed again below. Also, a number of new papers have been published that are relevant to this study (see below). The paper would benefit from referring to those as listed below.

5  Introduction line 24: The authors agree that SAI is not a good abbreviation, but they don't change it because it is wildly used. I do not agree that this is a good reason for using a misleading abbreviation. Again, the abbreviation should be changed to something like stratospheric aerosol geoengineering.

Indeed we do not see this as a major issue, stratospheric aerosol injection can be interpreted in a broad sense. However We changed the term stratospheric aerosol injection and its abbreviation SAI to stratospheric aerosol geoengineering and the

10  abbreviation SAG everywhere in the manuscript.

Page 2, Line 4: "stratospheric aerosol injection" is not correct if the authors are referring to Heckendorn, Niemeier, and English papers.

See previous response.

Page 3, Line 13: A reference to Richter et al. (2017) would be relevant.

15  We now also refer to Richter et al. (2017).

Page 3, Line 21: Please add a reference to MacMartin et al. (2017), who show that is is more efficient to inject at multiple injection locations outside the equator that at the equator.

We added a reference to MacMartin et al. (2017).

Page 7, Line 30: "so the exact ozone feedback is unclear" please remove that sentence and refer for example to Tilmes et al.

20  (2009), who discussed in detail the changes of stratospheric ozone as the result of stratospheric aerosol geoengineering.

We removed the sentence and added the following one: "Tilmes et al. (2009) also found considerable stratospheric ozone depletion in model experiments of SAG."

Page 9, Line 10ff: Here, you can also refer to Kravitz et al. (2017), who also looked at the same quantities.

We added the following sentence: "Kravitz et al. (2017) recently published a modelling study of SAG with $SO_2$ injection rates of up to $25\,\mathrm{Tg\,S\,yr^{-1}}$ and net forcing efficiency results similar to Niemeier and Timmreck (2015). But just as in our experiments, their individual SW and LW forcing efficiencies were considerably larger than those found by Niemeier and Timmreck (2015), indicating differences in the aerosol radiative effects between models."

Page 10, Line 18: Please also refer to Tilmes et al. (2017), who found very similar results for high vs. low injection cases.

We added the following sentence: "This is in agreement with a recent study by Tilmes et al. (2017), who also found that for large equatorial $SO_2$ injections the effectiveness does not increase for injections at higher altitude."

Page 13, line 19ff, and page 15, line 20: A reference to Richter et al. (2017) would be relevant, as well as mentioning difference to the results found in their study.

Page 13, Line 25: Please note, interactive chemistry has been shown to influence the response of geoengineering to the QBO (Richter et al., 2017).

We added the following sentence: "It has to be noted that the response of the QBO to SAG may be less severe when using interactive chemistry (Richter et al., 2017), probably due to ozone depletion resulting in a smaller stratospheric temperature anomaly."

**Summary of changes made in the revised manuscript**

The changes in the manuscript resulting from the comments by the Referee are marked on the following pages.

**References**

[revised manuscript text omitted]